



# Tracking daily NO$_x$ emissions from an urban agglomeration based on TROPOMI NO$_2$ and a local ensemble transform Kalman filter

Yawen Kong[1,2], Bo Zheng[3,4], Yuxi Liu[5]

[1]Ministry of Education Key Laboratory for Earth System Modeling, Department of Earth System Science, Tsinghua University, Beijing 100084, China
[2]State Key Laboratory of Remote Sensing Science, Aerospace Information Research Institute, Chinese Academy of Sciences, Beijing 100101, China
[3]Institute of Environment and Ecology, Tsinghua Shenzhen International Graduate School, Tsinghua University, Shenzhen 518055, China
[4]State Environmental Protection Key Laboratory of Sources and Control of Air Pollution Complex, Beijing 100084, China
[5]State Environmental Protection Key Laboratory of Environmental Pollution and Greenhouse Gases Co-control, Chinese Academy of Environmental Planning, Beijing 100041, China

*Correspondence to*: Yawen Kong (kongyw@aircas.ac.cn)

**Abstract.** Accurate, timely, and high-resolution NO$_x$ emissions are essential for formulating pollution control strategies and improving the accuracy of air quality modelling at fine scales. Since late 2018, the Tropospheric Monitoring Instrument (TROPOMI) aboard the Sentinel-5 Precursor (S5P) satellite has provided daily monitoring of NO$_2$ column concentrations with global coverage and a small footprint of 5.5 km × 3.5 km, offering great potential for tracking daily high-resolution NO$_x$ emissions. In this study, we develop a data assimilation and emission inversion framework that couples an Ensemble Kalman Filter with the Community Multiscale Air Quality model (CMAQ), to estimate daily NO$_x$ emissions at 3-km scales in Beijing and surrounding areas in 2020. By assimilating the TROPOMI NO$_2$ tropospheric vertical column densities (TVCDs) and taking the bottom-up inventory as prior emissions, we produce a posterior NO$_x$ emission dataset with a reasonable spatial distribution and daily variations at the 3-km scale. The proxy-based bottom-up emission mapping method at fine scales overestimates NO$_x$ emissions in densely populated urban areas, whereas our posterior emissions improve this mapping by reducing the overestimation of urban emissions and increasing emissions in rural areas. The posterior NO$_x$ emissions show considerable seasonal variations and provide more timely insight into NO$_x$ emission fluctuations, such as those caused by the COVID-19 lockdown measures. Evaluations using the TROPOMI NO$_2$ column retrievals and ground-based observations demonstrate that the posterior emissions substantially improved the accuracy of 3-km CMAQ simulations of the NO$_2$ TVCDs, as well as the daily surface NO$_2$ and O$_3$ concentrations in 2020. However, during the summer, despite notable improvements in surface NO$_2$ and O$_3$ simulations, positive biases in the posterior model simulations persist, indicating weaker constraints on surface emissions from satellite NO$_2$ column retrievals in summer. The posterior daily emissions on the 3-km scale estimated by our inversion system not only provide insights into the fine-scale emission dynamic patterns but also improve air quality modelling on the kilometer scale.



## 1 Introduction

As crucial atmospheric pollutants, nitrogen oxides ($NO_x = NO + NO_2$) play a vital role in tropospheric chemistry. $NO_x$ acts as an important precursor to ozone and secondary inorganic aerosols, which pose risks to the ecosystem and human health. $NO_x$ is primarily emitted into the atmosphere through anthropogenic sources such as industrial processes, transportation, and power generation(Li et al., 2017), as well as natural processes including soil emissions, lightning, and biomass burning. Accurate and up-to-date $NO_x$ emissions are of significant importance for formulating pollution control strategies and improving the precision

of air quality simulations.

Bottom-up emission inventories are used to characterize the spatiotemporal distribution of emissions and serve as the foundational input data for atmospheric chemical transport models (CTMs). Bottom-up approaches rely on activity data and average emission factors for each source sector, which could introduce uncertainties due to incomplete statistical data and misrepresentation of emission factors. Additionally, due to the delayed availability of activity data, bottom-up inventories

typically exhibit a temporal latency and cannot represent emissions accurately at the time of occurrence(Streets et al., 2003). Besides, bottom-up emission inventories typically allocate regional total emissions into fine-scale grids based on spatial proxies such as population densities, nighttime lights, road networks, and industrial gross domestic product (GDP) (Streets et al., 2003; Zhang et al., 2009; Geng et al., 2017). Such a proxy-based method assumes that emissions and spatial proxies are linearly correlated in space, which introduces biases at fine spatial scales (Wu et al., 2021). For instance, studies have shown

that proxy-based emission inventories tend to overestimate emissions in densely populated urban areas and underestimate emissions in suburban and rural areas, thus propagating uncertainties to high-resolution CTM simulations(Zheng et al., 2021b).

To improve $NO_x$ emission inventories, top-down approaches that derive $NO_x$ emissions from $NO_2$ observations have been developed and widely employed in literature to account for emission spatiotemporal variations (Streets et al., 2013; Martin et al., 2003; Miyazaki et al., 2017; Mijling and Van Der A, 2012; Ding et al., 2015; Kong et al., 2019), especially regarding

socioeconomic events such as the substantial emission fluctuations induced by the COVID-19 lockdowns (Ding et al., 2020; Miyazaki et al., 2020a; Kang et al., 2022; Feng et al., 2020; Zheng et al., 2020). The mass balance-based inversion method and its variants, which assume a direct linear relationship between local $NO_x$ emissions and $NO_2$ tropospheric columns(Martin et al., 2003), or between changes in $NO_x$ emissions and changes in $NO_2$ columns(Lamsal et al., 2011), have been widely adopted to infer regional $NO_x$ emissions from satellite $NO_2$ retrievals(Zheng et al., 2020; Zhao and Wang, 2009; Zhu et al., 2022;

Cooper et al., 2017). Although this method is suitable for short-lived gases and performs well at a coarse model resolution of tens of kilometers, the insufficient consideration of atmospheric transport between model grids and the nonlinear chemistry within each grid could cause emission-smearing errors(Cooper et al., 2017; Streets et al., 2013), particularly at fine sclaes.

The data assimilation technique is an alternate inversion method that works appropriately at fine spatial resolutions. The two common assimilation algorithms used for $NO_x$ emission inversion are four-dimensional variational assimilation (4D-VAR)

(Cooper et al., 2020; Wang et al., 2020; Qu et al., 2017; Qu et al., 2019) and Ensemble Kalman filter (EnKF) (Miyazaki et al.,



2020b). The target parameters (i.e., emissions) are optimized during data assimilation to minimize the discrepancy between model-predicted and observed pollutant concentrations. Atmospheric transport is accounted for by considering observations from the surrounding areas and their response to the target grid's emissions, and complicated atmospheric chemistry is incorporated into the model. Such methods are not limited to short-lived species like $NO_x$ and are also effective for long-lived

gases such as $CO_2$ and $CH_4$ (Kong et al., 2022; Liu et al., 2021; Pendergrass et al., 2023). However, due to the complex chemical simulation processes combined with the assimilation framework configurations, which require huge computational resources, studies on $NO_x$ inversions are often limited to short periods, monthly emission inversions, or coarse grid resolutions.

The Sentinel-5P Tropospheric Monitoring Instrument (TROPOMI), launched in October 2017, provides daily global coverage with a high-resolution footprint of $3.5 \times 5.5$ km$^2$ ($7 \times 3.5$ km$^2$ before 6 August 2019) and a high signal-to-noise ratio. Despite

low biases in early versions of TROPOMI $NO_2$ data (i.e., v1.2.x and v1.3.x) (Verhoelst et al., 2021; Lambert et al., 2021), the quality of satellite data has been improved through regular data validation and retrieval algorithm improvements including a major update in the FRESCO cloud retrieval and other algorithm updates(Van Geffen et al., 2022). TROPOMI provides global detailed $NO_2$ observations on a near-daily basis, which have been used extensively in emissions inversion across spatial scales (Van Geffen et al., 2022; Douros et al., 2023). In TROPOMI-based emission inversion studies, the primary focus is on point

sources or the total emissions from cities (Beirle et al., 2019; Wu et al., 2021; Lorente et al., 2019; Goldberg et al., 2019; Zhang et al., 2023). Recently, TROPOMI $NO_2$ retrievals have also been used to monitor emission reductions during the COVID-19 lockdowns (Zheng et al., 2020; Ding et al., 2020; Miyazaki et al., 2020a; Kang et al., 2022). These studies mainly focused on the early months of 2020 with a spatial resolution greater than several tens of kilometers, which is notably coarser than the fine footprint of TROPOMI. Furthermore, improvements to the model simulations are rarely considered in these studies.

By now, there is limited research on daily, high-resolution anthropogenic $NO_x$ emission inversion using TROPOMI retrievals, and there are very few reports on improving CTM simulations at the kilometer scale through $NO_x$ emission inversions. Herein, our study focuses on the daily anthropogenic $NO_x$ emission inversion on a 3-km resolution, which is close to the size of the TROPOMI pixel size, through an advanced data assimilation framework that couples an Ensemble Kalman Filter (EnKF) with the Community Multiscale Air Quality model (CMAQ) nested model. The main objective of this study is to understand the

spatial distribution patterns and the daily variations of anthropogenic $NO_x$ emissions at fine spatial resolutions, providing insights into the evaluation of the spatiotemporal distribution of bottom-up emission inventories. The models, data inputs, and observations used for the evaluations are detailed in Sect. 2. In Sect. 3, we analyse and evaluate the emission inversion results. Section 4 discusses inversion sensitivities and uncertainties. In Sect. 5, we provide a summary of the findings from this study.

## 2 Data and method

The emission inversion system framework (Fig. 1) is similar to our previous study for carbon flux inversion (Kong et al., 2022). In contrast to that study, which focused on a global scale, this research utilizes the regional CMAQ model for emission





inversion at daily, 3-km scales, based on high-resolution TROPOMI satellite NO$_2$ retrievals. The CMAQ model settings, satellite observation constraints, system configurations, as well as evaluation data, are described in Sect. 2.1-Sect. 2.6.

## 2.1 CMAQ model and a priori emissions

In the NO$_x$ inversion system, we used the CMAQv5.2.1 model to establish the relationship between emissions and simulated NO$_2$ concentrations. The meteorological fields that drive the CMAQ model are provided by the Weather Research and Forecasting Model (WRF) version 3.9.1. The initial and boundary conditions for WRF are derived from the National Centers for Environmental Prediction Final Analysis (NCEP-FNL) reanalysis data. The parameterization schemes for the WRF and CMAQ models are consistent with previous studies (Cheng et al., 2019; Zhang et al., 2019; Geng et al., 2021). For the CMAQ

chemical schemes, we used CB05 as the gas-phase chemical mechanism and AERO6 as the particulate matter chemical mechanism. CMAQ has 28 vertical levels in simulation, which are collapsed from the 46 sigma levels of the WRF model.

The target region of this study is the city of Beijing and its surrounding areas, which are highly susceptible to air pollution due to substantial anthropogenic emissions. To perform high-resolution simulations in this region, we designed three nested domains for the CMAQ model (Fig. S1). The first domain (D01) covers China and parts of surrounding countries with a

horizontal resolution of 27 km × 27 km; the second domain (D02) includes parts of northern and eastern China with a horizontal resolution of 9 km × 9 km, and the third domain (D03) covers Beijing city and its surrounding regions, including the Tianjin city and parts of Hebei Province, with a horizontal resolution of 3km × 3km. Since only the third domain is the target area for the emission inversion, the simulations over the first and second domains were performed before the inversion experiments to provide boundary conditions for the third domain. Since the inner nested domain is less influenced by the chemical processes

outside the first domain, the default background profiles are used as the boundary conditions for the D01 domain.

Anthropogenic emissions for mainland China in 2020 are obtained from the Multiresolution Emission Inventory in China (MEIC, http://meicmodel.org/) (Zheng et al., 2018; Li et al., 2017), which is dynamically updated using a bottom-up approach based on near-real-time activity indicators (Zheng et al., 2021a). The MEIC emission inventory is spatially and temporally allocated to match the CMAQ model domain using spatial proxies and empirical temporal profiles. Emissions from the regions

outside mainland China are obtained from the MIX emission inventory (Li et al., 2017b), which is coupled with the MEIC emission inventory over the first domain. The biogenic emissions are dynamically generated by the Model of Emissions of Gases and Aerosols from Nature (MEGAN v2.1.4), which is driven by the WRF model over the same CMAQ domains.

## 2.2 TROPOMI NO$_2$ retrievals

TROPOMI is an advanced spectrometer on board the European Space Agency's (ESA) Sentinel 5 Precursor (S5P) satellite,

with the Equator crossing time at approximately 13:30 local time (LT). TROPOMI provides measurements in a wide spectral range covering ultraviolet, visible, and shortwave infrared wavelengths, and the visible band (400-496 nm) is used for the NO$_2$ retrievals. TROPOMI has great advantages in monitoring and inferring trace gas emissions at fine spatial scales, as it provides



near-daily global coverage with a wide swath of 2600 km, and an unprecedented high-resolution footprint. A negative bias was identified in early versions of the TROPOMI $NO_2$ data (i.e., v1.2.x and v1.3.x), particularly in polluted regions, compared to

ground-based or OMI measurements (Verhoelst et al., 2021; Lambert et al., 2021). With a major update in the FRESCO cloud retrieval and other algorithm updates including an adjustment of the surface albedo (since v2.2) (Van Geffen et al., 2022), new versions (since v1.4) of the TROPOMI $NO_2$ products have substantially reduced the biases. In this study, we used version 2.3.1 of the TROPOMI $NO_2$ based on the v2.3.1 processor developed and maintained by the Royal Netherlands Meteorological Institute (KNMI), which has been adopted in recent $NO_x$ inversion studies (Zhang et al., 2023; Li et al., 2023).

Before assimilating the TROPOMI $NO_2$ tropospheric vertical column density (TVCD) retrievals, we first selected the pixels with a quality assurance value greater than 0.5 to ensure data quality for assimilation and model comparison purposes, as recommended by the product user manual (Eskes et al., 2022). We further excluded the pixels with cloud fraction exceeding 40% to reduce retrieval errors. For comparison with the CMAQ model, we used the area-weighted average method to convert the pixel-specific satellite retrievals into a gridded data product to match the resolution of the CMAQ grid cell (3 km × 3 km).

**2.3 Local ensemble transform Kalman Filter**

The core algorithm for our inversion system is the local ensemble transform Kalman Filter (LETKF) (Hunt et al., 2007), which is a variant of the Ensemble Kalman Filter (EnKF) approach. The LETKF is designed for ease of implementation and computational efficiency compared to traditional EnKF approaches and adjoint-based variational methods (e.g., 4D-Var). First, in LETKF, the analysis state can be solved independently at each model grid, and only the observations within a specified local

area around each model grid are assimilated. In addition, LETKF uses an explicit localization scheme in both space and time, which ensures the accuracy and efficiency of the inversion process even with a limited size of ensemble members, particularly in regions with sufficient observations. The advantages of the LETKF method make it applicable to high-dimensional chemical assimilation systems.

Similar to the EnKF-based methods, LETKF uses the ensemble members of the prior states $x_i^b$ ($i = 1,2,...,k$) to approximate

the a priori error covariance matrix, where $k$ is the ensemble size. In this study, the state vector $x$ refers to the scaling factors of the $NO_x$ emissions in each model grid cell. Research (e.g., (Inness et al., 2015)) has shown that optimizing the initial $NO_2$ concentrations by assimilating the satellite observations has minimal impact on the CTM simulations due to the short lifetime of $NO_2$, so we only adjust the emissions in the inversion system. Since emissions are not the variables simulated by the CMAQ model, we use a simple dynamic model to forecast the ensemble mean of the prior states in each assimilation step ($t$):

$$\overline{x}_t^b = (\sum_1^l \overline{x}_{t-l}^a + 1)/3 \tag{1}$$

Where superscripts $a$ and $b$ refer to the posterior and the prior state, respectively, $\overline{x}$ is the ensemble mean of the state vector, $t$ refers to the time step of the assimilation system, and $l = 2$. With the dynamic model, the optimized information from the two previous time steps can be propagated to the current state, allowing for achieving assimilation balance more rapidly (Peters et



al., 2007). The ensemble of perturbation states $\mathbf{X}^b$ was generated through Cholesky decomposition to a prior error covariance

matrix $\mathbf{P}^b$ (i.e., $\mathbf{P}^b = \mathbf{X}^b(\mathbf{X}^b)^\mathrm{T}/(k-1)$). Then the ensemble members $\boldsymbol{x}_i^b (i = 1,2,\ldots,k)$ were constructed by adding the ensemble

mean $\overline{\boldsymbol{x}}^b$ to the $i$th column of $\boldsymbol{X}^b$.

In each assimilation step, the state vector can be updated using the following equations:

$$\overline{\boldsymbol{x}}^a = \overline{\boldsymbol{x}}^b + \mathbf{X}^b\,\overline{\boldsymbol{w}}^a \tag{2}$$

$$\mathbf{X}^a = \mathbf{X}^b[(k-1)\widetilde{\mathbf{P}}^a]^{1/2} \tag{3}$$


$$\overline{\boldsymbol{w}}^a = \widetilde{\mathbf{P}}^a(\mathbf{Y}^b)^\mathrm{T}\mathbf{R}^{-1}(\boldsymbol{y}^o - \overline{\boldsymbol{y}}^b) \tag{4}$$

$$\widetilde{\mathbf{P}}^a = [(k-1)\mathbf{I} + (\mathbf{Y}^b)^\mathrm{T}\mathbf{R}^{-1}\mathbf{Y}^b]^{-1} \tag{5}$$

Where $\boldsymbol{y}^o$ refers to the assimilated satellite observations, $\overline{\boldsymbol{y}}^b$ is the ensemble mean of the model simulated observations $\boldsymbol{y}^{b(i)}(i$

= 1,2,…,$k$) which are mapped to the observation space through the observation operator $\boldsymbol{y}^b = H(\boldsymbol{x}^b)$, $\mathbf{Y}^b$ is the ensemble

perturbation matrix whose $i$th column represents $\boldsymbol{y}^{b(i)} - \overline{\boldsymbol{y}}^b$ ($i = 1,2,\ldots,k$), $\widetilde{\mathbf{P}}^a$ is the analysis error covariance matrix in the

ensemble space, $\mathbf{R}$ represents observation error covariance, $\overline{\boldsymbol{w}}^a$ is the mean weighting vectors that specify what linear

combinations of the ensemble perturbations ($\boldsymbol{X}^b$) to be added to the prior ensemble mean ($\overline{\boldsymbol{x}}^b$)to form the posterior ensemble

mean ($\overline{\boldsymbol{x}}^a$), $\mathbf{I}$ is the identity matrix.

## 2.4 Inversion system setup

Because of the notable emission fluctuations during the COVID-19 pandemic in 2020, which could be difficult to accurately

characterize by the bottom-up emission inventory, we select the period of the inversion experiments from December 2019 to

the end of December 2020 to monitor the daily emission changes constrained by the TROPOMI satellite measurements. The

ensemble size is set to 60 in our inversion system which is reasonable since LETKF has been shown to perform well even with

a small ensemble size (Miyoshi and Yamane, 2007). The uncertainty for the prior emissions is prescribed as 100%, and for

each assimilation cycle, the prior error covariance maintains a fixed uncertainty value to prevent filter divergence, which is

similar to our previous study (Kong et al., 2022). The observation localization is implemented by multiplying the observation

error covariance matrix by the inverse of a localization weighting function $\exp(-r^2/L^2)$ to gradually reduce the effect of

observations as the distance from the analysis grid increases. Here $r$ denotes the distance between the observations and the

analysis grid point, and $L$ denotes the localization radius parameter. The observations beyond the localization radius ($L$) do not

impact the emissions at the analysis grid cell in the inversion system. Considering the typical lifetime of $NO_2$ and wind speed

around Beijing (Wu et al. (2021)), we initially set the localization parameter ($L$) to 36 km. We also conduct two sensitivity

experiments (Sect. 2.6) using 3 km and 81 km as the localization parameter to analyse their impacts on the emission inversions.



To minimize the influence of the lack of observational constraints over the model domain, we exclude the days with satellite coverage below 70% of the total grids across the entire study area from the inversion process. In such cases, the corresponding posterior emissions on those dates are supplemented by the average of emissions from the adjacent days. Fig. S2 shows the

number of days with coverage exceeding 70% per month, as well as the corresponding data coverage histogram. Generally, satellite coverage is adequate throughout most of the year, but during the rainy season (i.e., July and August) in North China, there are fewer valid satellite retrievals, with only 35% and 32% of days, respectively, satisfying the coverage requirements.

The standard TROPOMI NO$_2$ TVCDs retrieval product from the Royal Netherlands Meteorological Institute (KNMI) is used as the observation constraint in our emission inversion system. The TROPOMI NO$_2$ retrievals are resampled to the model 3-

km grid using an area-weighted average method. The model simulated NO$_2$ column is calculated by multiplying the CMAQ model partial column profile with the tropospheric averaging kernel (Douros et al., 2023; Eskes et al., 2022):

$$V_m = \sum_l A_l^{\text{trop}} x_l^m, \; l \le l_{\text{trop}} \tag{6}$$

Where $V_m$ represents the model simulated NO$_2$ TVCDs, $x_l^m$ represents the model partial column in layer $l$, $A_l^{\text{trop}}$ is the tropospheric averaging kernel (AK) which describes the vertical sensitivity of the satellite instrument to NO$_2$ concentrations

at different vertical layers. Here the averaging kernel is interpolated to the model layers. The AK-based method removes the dependence of the model-satellite comparison on the a priori profile of the global TM5-MP model used in the TROPOMI satellite retrievals (Eskes and Boersma, 2003). In the sensitivity experiments (Sect. 2.6), we further implement two additional schemes for the model-satellite comparisons to analyse the impact of the a priori profile and AK on the emission inversions.

### 2.5 Sensitivity experiments

We designed six sensitivity inversion experiments to evaluate our inversion system (Table 1). The experiments Exp1-Exp4 are designed to analyse the impact of the a priori profile and AK on emission inversions. Exp_L3km and Exp_L81km were designed to investigate the effect of different observation localization radius parameters on the emission inversion.

The retrieval of the tropospheric NO$_2$ TVCD is substantially influenced by the assumed a-priori NO$_2$ profiles, which are used in the calculation of the air mass factor (AMF) that converts slant column into vertical column (Cooper et al., 2020; Huijnen

et al., 2010; Douros et al., 2023). Due to the differences in the shape of the satellite and model a priori profiles, the satellite-model comparison is influenced by the shape of the a priori profile. There are two common ways to eliminate the impact of the a priori profiles on model-satellite comparisons: applying satellite averaging kernels to modelled partial columns or developing new satellite retrievals with modelled profiles instead of using the a priori profiles derived from the TM5-MP model (Douros et al., 2023; Eskes et al., 2022). The former method is the approach adopted in our study (i.e., Exp1 in Table1),

while the latter is used for sensitivity inversions (i.e., Exp2-Exp3 in Table1). We used the posterior emission (derived from Exp 1)-driven CMAQ profiles in Exp2 and the prior emission-driven CMAQ profiles in Exp3 to recalculate the satellite NO$_2$ TVCDs product. The new profiles have a horizontal resolution of 3 km over the D03 domain of the CMAQ model. The new



TROPOMI NO₂ retrievals $V_{o,new}$ for Exp2-Exp3 are derived from the new a-priori profiles from CMAQ simulations as follows (Douros et al., 2023; Eskes et al., 2022):

$$V_{o,new} = \frac{M^{trop}}{M^{trop}_{cmaq}} V_o \qquad (7)$$

$$M^{trop}(x_m) = \frac{M^{trop}(x_a) \sum_l A^{trop}_l x^m_l}{\sum_l x^m_l} \qquad (8)$$

Where $M^{trop}(x_m)$ is a new tropospheric air mass factor computed with the CMAQ model profile, $M^{trop}(x_a)$ is the air mass factor depending on the TM5-MP model a priori profile, $V_o$ is the standard TROPOMI NO₂ retrieval. In Exp2 and Exp3, $x_m$ represents the posterior-emission-driven CMAQ profiles and the prior-emission-driven CMAQ profiles, respectively. For Exp2 and Exp3, the modelled NO₂ TVCDs are calculated by directly integrating the partial NO₂ columns within the troposphere without applying satellite AK (CMAQ-w/o-A). Additionally, we include another experiment (Exp4) to compare the standard TROPOMI products with the model columns directly without applying AK. It should be noted that the comparison in Exp4 represents the influence from the a priori profile of TROPOMI retrievals.

**2.6 Evaluation observations**

The ground-based observations are used to evaluate the impact of the emission inversion constrained by the TROPOMI satellite retrievals on the improvement of the CMAQ simulations at the 3 km scale. The surface observations are obtained from the monitoring stations maintained by China National Environmental Monitoring Center (CNEMC, http://www.cnemc.cn/, last access: 20 January 2024 ). NO₂ and O₃ measurements are used for inversion model evaluation in our study since NO₂ and O₃ concentrations are highly sensitive to the adjustment of NO$_x$ emissions.

**3 Results**

**3.1 Evaluation of CMAQ simulations driven by posterior emissions**

The CMAQ simulations driven by the proxy-based prior emission inventory are broadly consistent with the TROPOMI NO₂ TVCDs (Fig. S3) and surface-based NO₂ measurement (Fig. S4) at the 27 km grid scale but overestimate NO₂ concentrations at the 3 km scale over emission hotspot areas (Figs. 2 and 3). Zheng et al. (2021b) indicated that CTM simulations driven by proxy-based emission maps perform better at coarse resolutions of tens of kilometers since coarse grids generally contain entire administrative units and are less affected by emission spatial allocation based on proxies (e.g., population), which tend to misallocate the industrial emissions located in suburban areas to densely populated city centers. In this section, we evaluate the effectiveness of our inversion system by comparing the CMAQ simulations at the 3 km scale driven by the prior and posterior NO$_x$ emissions, respectively, against TROPOMI NO₂ retrievals and the independent ground-based observations.



### 3.1.1 NO₂ tropospheric columns

The CMAQ simulations, driven by the posterior emissions, substantially narrowed the discrepancies between simulated NO₂ TVCDs and TROPOMI NO₂ retrievals (Fig. 2). In the simulations driven by prior emissions, the grid cells exhibiting a positive bias were more prevalent over the whole region, while the posterior simulation effectively correct this bias. The Root Mean Squared Error (RMSE) values decreased from 14.25 in winter, 3.52 in spring, 3.37 in summer, and 6.34 in autumn in the prior simulation to 5.94, 1.44, 1.02, and 1.9, respectively, in the posterior simulation (all values $\times 10^{15}$ molecules cm$^{-2}$).

Higher NO₂ TVCDs are typically observed over urban areas characterized by high population densities and extensive road networks. Using these factors as spatial proxies in emissions mapping is thus justified to a certain degree. However, the NO₂ TVCDs from prior simulations indicate substantial overestimations in urban environments across various seasons, particularly in the central districts of Beijing, Tianjin, and Baoding. This suggests that the prior emission inventory might have overestimated emissions in these areas. In contrast, the NO₂ TVCDs from posterior simulations show a considerable reduction in these overestimation errors when compared to the TROPOMI satellite observations, aligning more closely with the observed spatial gradient distribution patterns. Figure 4 illustrates the spatial maps of both prior and posterior NO$_x$ emissions, highlighting the areas that contribute to the improvement observed in the simulated NO₂ concentration distributions.

The improvements in the simulated NO₂ TVCDs are more pronounced in spring and summer compared to autumn and winter. In the warmer months, due to NO₂'s short atmospheric lifetime, its concentrations are less influenced by long-range transport, leading to high-concentration regions being more localized around emission hotspots. The NO₂ TVCDs simulations in these seasons thus greatly benefit from our inversion-optimized emissions. In the cooler months, the prior simulations overestimate NO₂ TVCDs across most high-emission areas, and the posterior simulations can also mitigate these overestimations. Despite these improvements, the posterior simulations still show overestimation errors in the southern region of our study area during the colder seasons. This might be attributed to the longer NO₂ lifetime in colder conditions, which increases its susceptibility to regional transport and accumulation. Specifically, pollution transport from sources south of Baoding City contributes to the overestimated NO₂ levels in this region. Our inversions at the 3 km scale were limited to optimizing emissions in the innermost domain and did not address emissions outside this area. Expanding the simulation domain could potentially rectify the persisting overestimation errors near Baoding City, although this would require additional computation resources.

### 3.1.2 Surface NO₂ and O₃ concentrations

As shown in Fig. 3, the daily NO₂ concentrations modelled with input from the bottom-up inventory are substantially overestimated, while simulations utilizing posterior NO$_x$ emissions have considerably improved the accuracy of the surface NO₂ simulations. Over the entire year of 2020, the RMSE between the modelled and observed NO₂ concentrations decreased from 38.93 µg/m³ in the prior emission-based simulations to 13.48 µg/m³ in the posterior simulations. However, it is important to note that while the posterior simulations closely matched observed concentrations, the improvement in NO₂ simulations during the summer was relatively limited. One possible reason for the large simulation biases in summer is the lack of satellite





observation constraints during the rainy season (Fig. S2a). Additionally, the AK describes the sensitivity of the satellite instrument to tracer concentrations at different altitudes and the sensitivity is small near the surface (Eskes and Boersma, 2003). This suggests that satellite retrievals of $NO_2$ columns offer weak constraints on ground emissions, especially in summer, as

indicated by smaller AK values near the surface compared to winter (Fig. S5), thereby resulting in limited improvements in NO2 simulations.

In terms of $O_3$ concentrations, the optimized $NO_x$ emissions also improve the simulation accuracy since $NO_x$ can influence the formation and depletion of $O_3$. The RMSE between the simulated $O_3$ concentrations and ground-based $O_3$ observations decreased from 26.11 µg/m³ to 13.91 µg/m³. The overestimation of the bottom-up $NO_x$ emissions leads to negative biases in

the $O_3$ simulations throughout the year. In summer, the posterior $O_3$ simulations match the observations well but are slightly below the observed values, which also indicates that the posterior $NO_x$ emissions may still be overestimated during the summer.

## 3.2 Spatial-temporal variations of the posterior $NO_x$ emissions

### 3.2.1 Spatial distribution of the $NO_x$ emissions

The posterior emissions have led to substantial improvements in the accuracy of the 3-km CMAQ simulations, which could

be attributed to the improved $NO_x$ emission distribution patterns after being constrained by satellite observations. Here, we first analyse the changes in the spatial distribution of posterior $NO_x$ emissions at the 3-km scales.

Fig. 4 shows the spatial map of the prior and posterior emissions for the four seasons in 2020. The prior emission maps show the highest intensity in urban centers, including Beijing, Baoding, and Tianjin urban areas. The satellite-constrained posterior emissions maintain a similar spatial distribution with the emission hotspots broadly consistent with the prior emission inventory.

However, the posterior emission maps substantially reduce emissions from city centers and reallocate these emissions to other areas, such as increasing the emissions from inter-city transportation, among other changes. From the spatial map of the total population as shown in Fig. S6, the grids with decreased emissions in the posterior emission maps are consistent with areas of high population density, especially in summer. The areas with high population density are mainly located in the city center. This indicates that at the kilometer-scale spatial resolution, emission inventories allocated based on the population density as

a spatial proxy tend to overestimate emissions in grid cells located in urban centers in Beijing and its surrounding areas.

The posterior $NO_x$ emissions for the year 2020 (657 kt $NO_x$) decreased by 23.7% compared to the prior inventory (861 kt $NO_x$). The largest reduction in emissions occurs in the winter and autumn, with a decline of 44.5% and 36.4%, respectively. The total emissions during the summer are comparable to the prior emission inventory. This is mainly due to the weaker reduction of emissions occurring in urban areas when compared to other seasons, coupled with a more pronounced increase in emissions

in non-urban areas, such as the road network, which leads to regional total emissions remaining at relatively high levels.





### 3.2.2 Temporal variations of the NO$_x$ emissions

Performing emission inversion on a daily basis can illustrate daily emission changes, which are difficult to capture in the prior emission inventory. The posterior emissions reflect the reduction and recovery in emissions in early 2020 (Fig. 5) driven by the implementation and relaxation of COVID-19 containment measures, as well as the impact of the Chinese Lunar New Year holiday. The reduction of NO$_2$ concentrations due to the pandemic lockdown measures endured from early 2020 to mid-March, which is consistent with the time series of the posterior emissions. However, the prior emission inventory only captures the emission reduction that occurred from early February to mid-February 2020, mainly caused by the Chinese Lunar New Year. The posterior emission estimates also indicate a period of emission reduction in mid-to-late June 2020, coinciding with the sudden outbreak of the epidemic and the subsequent lockdown measures implemented at the Xinfadi market in Beijing, China.

The satellite observations introduce considerable seasonal variations in the posterior NO$_x$ emissions, while the prior NO$_x$ emission inventory does not exhibit obvious seasonal variations. In winter, emissions are overestimated, mainly due to the overestimation of urban emissions (Fig. 4). In summer, total emissions are comparable to the posterior emissions because the emission reduction in densely populated urban centers is compensated for by an increase in emissions from other regions. The seasonal variation of the posterior NO$_x$ emission estimate in our research is similar to the results obtained by previous studies (Wang et al., 2007; Qu et al., 2017; Miyazaki et al., 2017). Qu et al. (2017) utilized OMI measurements to infer the NO$_x$ emissions in China, and the seasonal pattern of NO$_x$ emissions for China and Beijing City is consistent with our study. Since we focus only on anthropogenic emissions in urban areas, this study does not include natural emissions from soil and lightning sources, except for the biogenic emissions which are dynamically generated by the MEGAN model. However, these natural sources are implicitly included in the posterior emissions. Previous studies have attributed the summer peak of NO$_x$ emissions to contributions from natural sources since high temperatures in summer lead to more emissions from soil and fertilization, and there is also a higher frequency of lightning.(Miyazaki and Eskes, 2013; Qu et al., 2017). We select grids where anthropogenic and biogenic sources account for more than 90% of the total emissions, respectively. The seasonal variations (Fig. 6) show that the grids dominated by biogenic sources contribute emissions mainly during the warm seasons, and the prior biogenic emissions are significantly underestimated.

### 4 Discussions

#### 4.1 Impact of the a-priori profile and averaging kernel on the emission inversion

Four experiments are designed to test the impacts of the a-priori profile and satellite AK on the NO$_x$ emission inversions, and the settings are described in Sect. 2.6 and Table 1. The daily posterior emission time series from the four sets of experiments are shown in Fig. 7. The results indicate that the emissions obtained from the four sets of experiments are generally consistent and robust throughout the year except for summer. First, the differences between Exp1, Exp2, and Exp3 are relatively small, indicating that the impacts of the two optimal satellite-model comparison methods on the inversions are nearly equivalent. In



contrast, the emissions obtained from Exp4 are slightly lower than those from Exp1 during the summer.

Due to the weaker AK sensitivity at altitudes below 2 km during summer (Fig. S5), Exp1, which uses AK in the model $NO_2$ column calculation, yields lower model columns in summer compared to Exp4. As a result, the positive bias in model columns

with satellite columns is smaller in Exp1, leading to higher emissions compared to Exp4. In winter, AK values are relatively higher than in summer and increase rapidly with altitude within 2 km. Consequently, the AK-based model columns are better at reflecting emissions in the lower atmosphere in winter than in summer. Both Exp2 and Exp3 use model-derived vertical column concentrations without using AK, which is similar to Exp4, but the emission inversion results are higher than Exp4 in summer. This is because although the model vertical columns are calculated using the same method for Exp4 in summer, the

new satellite products calculated using CMAQ-based a-priori profiles show higher columns during the summer compared to the standard satellite product, especially in emission hotspots (Fig. 8). Therefore, the posterior emissions are higher than those of Exp4 in summer. Douros et al. (2023) also produced a new TROPOMI product over Europe using the modelled profiles from the high-resolution regional model (0.1°×0.1°), and they also found an increase in the tropospheric columns on emission hotspots over the summer months.

The comparison between Exp2 and Exp3 shows similar emission inversion results. This is because the two new satellite products are consistent in the values (Fig. S7), which are calculated based on the CMAQ profiles simulated by the prior and posterior emissions, respectively. This indicates that the vertical profile shapes of the modelled NO2 concentrations driven by both prior and posterior emission inventories are similar, provided that the model and model resolution settings are consistent. Therefore, when utilizing newly generated satellite products for assimilation and inversion, it may not be necessary to update

the a priori profiles in each assimilation step, and the model profiles simulated using the prior emissions can be used to calculate new satellite products before the inversion process. This result is similar to the study by Cooper et al. (2020), which performed experiments using synthetic observations to test the a-priori profile on the emission inversions. Their study found that the inversion error is the lowest when the a priori profiles are updated during the assimilation process, but using the profile from the initial model state without updating during the assimilation can also get accurate inversions because their profile shapes

are similar, regardless of whether they are updated.

**4.2 Impact of the observation localization parameters on the emission inversion**

The observation localization of LETKF is implemented by assimilating observations within a specified local area and using a decay function to reduce the impact of observations as the distance from the analysis grid increases. To evaluate the impact of different L values on the $NO_x$ emission inversions, we perform two additional experiments with $L = 3$ km (Exp_L3km) and $L$

$= 81$ km (Exp_L81km), respectively. The experiments were conducted for the periods of December 2019 and June 2020, and then the posterior $NO_2$ simulations were evaluated through comparisons with TROPOMI $NO_2$ TVCDs (Fig. 9). The results showed that both the Exp_L3km and Exp_L81km can optimize $NO_x$ emissions to some extent, but the effectiveness was the weakest for Exp_L81km, followed by Exp_L3km, while the localization parameter of 36 km produced the best performance.





This may be because the 81 km × 81 km area includes a large number of observations which could disturb the optimization of the emissions in the target grid. On the other hand, observations at longer distances were already outside the influence area of the emission sources from the target grid. The area with a 3 km radius contains too few observations and has insufficient responsiveness to the transport of emissions from the target grid. In practice, the selection of an appropriate localization parameter may require extensive tuning or adaptive adjustments based on meteorological conditions, but this will impose huge computational burdens. Therefore, we tend to choose a theoretically reasonable parameter or observation localization.

**4.3 Future improvement directions**

Uncertainties in the $NO_x$ emission inversion partly result from inaccurate depictions of the chemical and physical mechanisms in the CTMs, errors in satellite retrievals, and limitations in the assimilation techniques. In terms of our emission inversion framework, several factors can introduce uncertainties to the inversions. Firstly, the settings of the CTM simulations, such as the configuration of the model layers, might be important for the depiction of the column concentrations where future improvements can be made. Secondly, the TROPOMI satellite's local overpass time in the min-afternoon means that the inversion system still cannot capture hourly variations in $NO_x$ emissions, and therefore the hourly allocations in the posterior emissions are the same as in the bottom-up inventory. Besides, the weak sensitivity of the satellite measurements to the $NO_2$ concentrations at lower altitudes leads to limited constraints on surface emissions, particularly during the summer (Miyazaki and Eskes, 2013). A recent study by He et al. (2022) suggests that the incorporation of hourly ground-based observations may help to improve surface emission inversions. In addition, how model simulations are matched to satellite column concentrations and the interpolation process for satellite-model comparison can also introduce uncertainties to the $NO_x$ emission inversion. Finally, although data assimilation methods take into account the influence of the observations outside of the target grid, the optimal influence distance is difficult to determine. An improved observation assimilation scheme, such as taking into account the correlation between the emissions in the target grid with the observations within the localization area may hold promise for improving the accuracy of emissions inversions.

**5. Conclusions**

The advantages of TROPOMI's small footprint and near-daily global coverage enable research on fine-scale emission inversions. In this study, we develop an advanced emission inversion system by combining the CMAQ nested model with the LETKF algorithm to assimilate TROPOMI $NO_2$ retrievals. We obtain the posterior $NO_x$ emission maps on a daily basis at 3 km scales in Beijing and surrounding areas. The posterior emissions offer a more reasonable spatial disaggregation and capture temporal changes effectively. In terms of spatial distribution, we further explain how proxy-based spatial allocation schemes in the prior emission inventory tend to overestimate urban emissions. In terms of temporal variations, the posterior emissions show considerable seasonal changes and provide timely insights into daily emission variations, such as the emission reductions induced by the COVID-19 control measures. We conduct sensitivity experiments to analyse the impact of the a-priori profiles

used in the satellite retrievals and the use of AKs for calculating model $NO_2$ columns on the inversion results. The experiments indicate the robustness of our inversion, with a greater sensitivity to different model-satellite comparison schemes in summer. Our posterior $NO_x$ emissions improve the accuracy of the CMAQ simulations of $NO_2$ and $O_3$ concentrations at 3-km scales for the whole year in 2020. However, in summer, despite notable improvements in the CMAQ simulations, a positive model bias persists, implying that TROPOMI $NO_2$ column retrievals provide weaker constraints on surface $NO_x$ emissions in summer.

Overall, the high-resolution $NO_x$ emission inversion framework developed in this study improves our understanding of the spatial distribution patterns and the daily variations of anthropogenic $NO_x$ emissions in fine scales, providing an instructive example for improving fine-scale modelling and prediction.

**Data availability.**

The inversion datasets generated in this study are available from the corresponding author upon reasonable request.

**Author contributions.**

YK conceived the study and performed the emission inversion experiments, carried out the analysis, and prepared the initial draft. ZB supervised the research and revised the manuscript. YL helped with the CMAQ simulations and the bottom-up emission preparation. All of the authors contributed to the writing and editing of the paper.

**Competing interests.**

The authors declare that they have no conflict of interest.

**Acknowledgements.**

This work was supported by the National Key R&D program (2023YFC3705601) and the National Natural Science Foundation of China (grant nos. 42275191 and 42207119). The S5P TROPOMI $NO_2$ data are downloaded from https://data-portal.s5p-pal.com/. We would like to acknowledge Professor Qiang Zhang of Tsinghua University for his invaluable comments and
support.

**Financial support.**

This research has been supported by the National Key R&D program (2023YFC3705601) and the National Natural Science Foundation of China (grant nos. 42275191 and 42207119).



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





**Table 1.** Parameters for the sensitivity inversion experiments.

| Inversion Experiment | TROPOMI retrieval | Modelled NO$_2$ TVCDs | Localization radius |
|---|---|---|---|
| Exp1 | TROPOMI-KNMI[a] | CMAQ-A[d] | 36 km |
| Exp2 | (TROPOMI-CMAQ-opti)[b] | CMAQ-w/o-A[e] | 36 km |
| Exp3 | (TROPOMI-CMAQ-base)[c] | CMAQ-w/o-A | 36 km |
| Exp4 | TROPOMI-KNMI | CMAQ-w/o-A | 36 km |
| Exp_L3km | TROPOMI-KNMI | Same as in Exp1 | 3 km |
| Exp_L81km | TROPOMI-KNMI | Same as in Exp1 | 81 km |

[a]TROPOMI-KNMI refers to the standard TROPOMI retrieval product from KNMI. [b]TROPOMI-CMAQ-opti refers to the new TROPOMI NO$_2$ retrievals using the posterior CMAQ simulation from Exp1 as the a-priori profiles. [c]TROPOMI-CMAQ-base refers to the new TROPOMI NO$_2$ retrievals using the prior CMAQ simulation from Exp1 as the a-priori profiles. [d]CMAQ-A represents CMAQ NO$_2$ TVCDs calculated using satellite Aks. [e]CMAQ-w/o-A represents CMAQ NO$_2$ TVCDs calculated without applying satellite Aks.

600



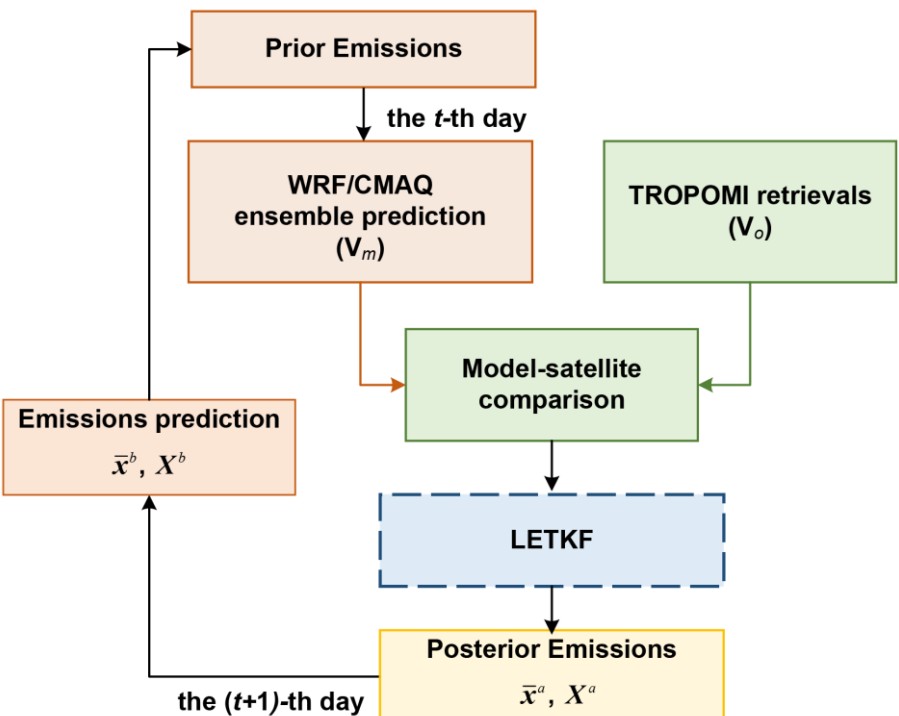

**Figure 1. Flowchart of the emission inversion system.**





**Figure 2.** Comparison of the spatial distributions of TROPOMI NO₂ TVCD retrievals (the left column) with CMAQ-modeled
NO₂ TVCDs driven by the prior (the second column) and posterior (the third column) emissions. The last column is the scatter
plot of the TROPOMI retrievals with the CMAQ-modeled NO₂ TVCDs simulated using the prior (blue) and posterior (red)
emissions. From top to bottom, the rows represent Winter (DJF), Spring (MAM), Summer (JJA) and Autumn (SON),
respectively.





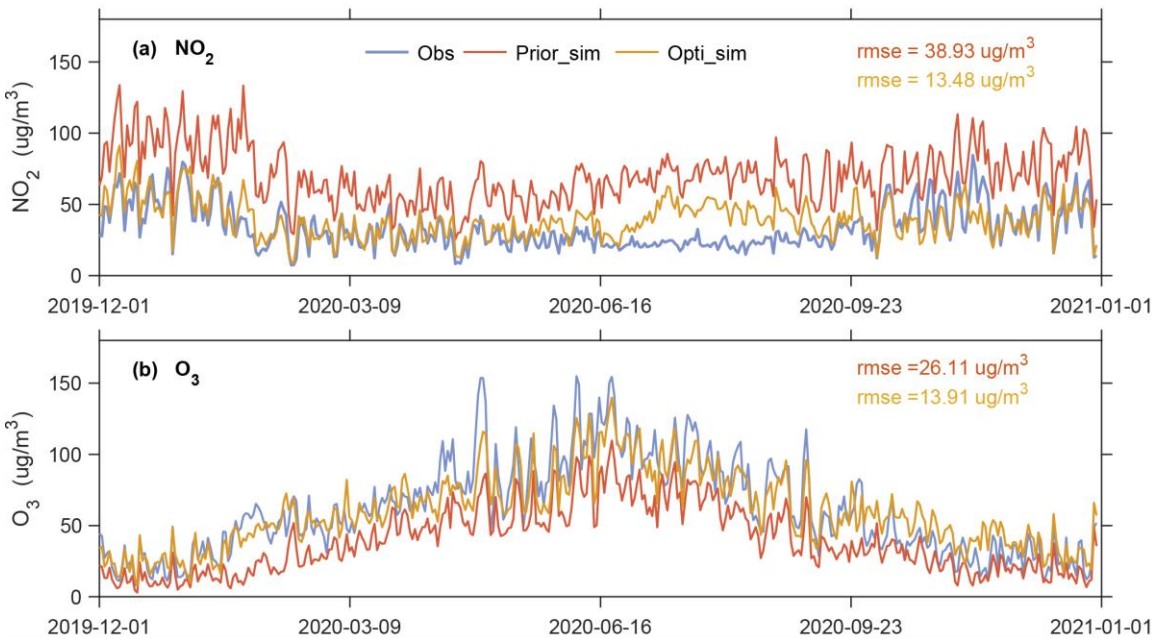

**Figure 3.** Comparison of the ground-based daily (a) NO$_2$ and (b) O$_3$ concentrations with the simulations utilizing prior and posterior NO$_x$ emissions as the model input.





**Figure 4.** The spatial map of the bottom-up and top-down emissions for the four seasons in 2020. The last column shows the differences between the two emissions. From top to bottom, the rows represent Winter (DJF), Spring (MAM), Summer (JJA) and Autumn (SON), respectively.



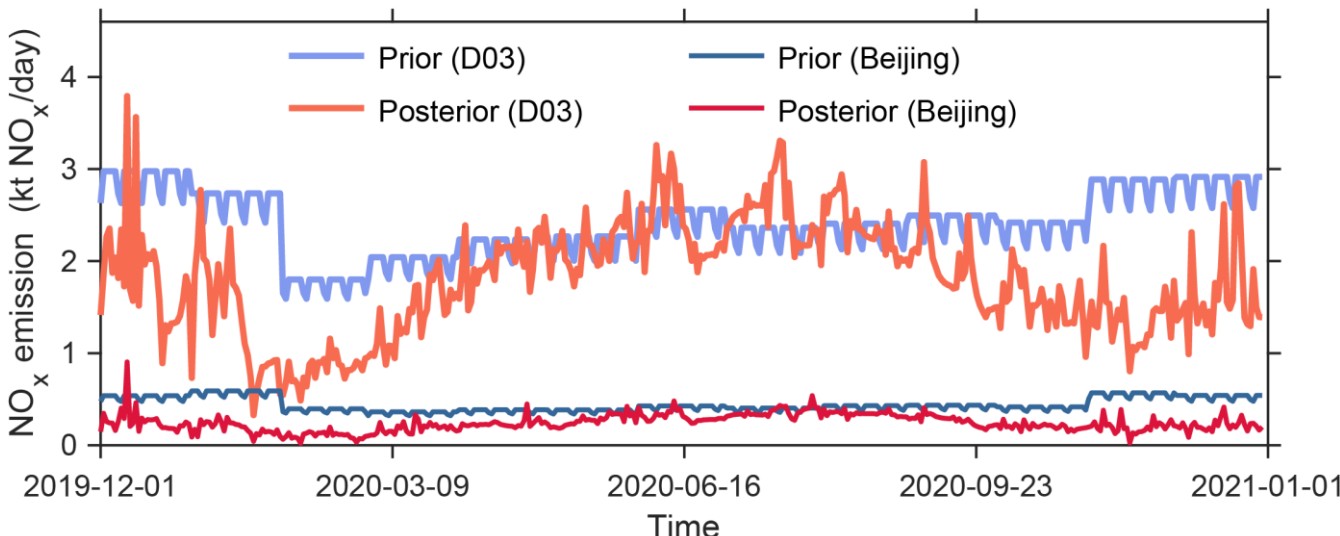

**Figure 5.** Time series of the bottom-up and top-down daily NO$_x$ emissions for (a) domain D03 and (b) Beijing City.



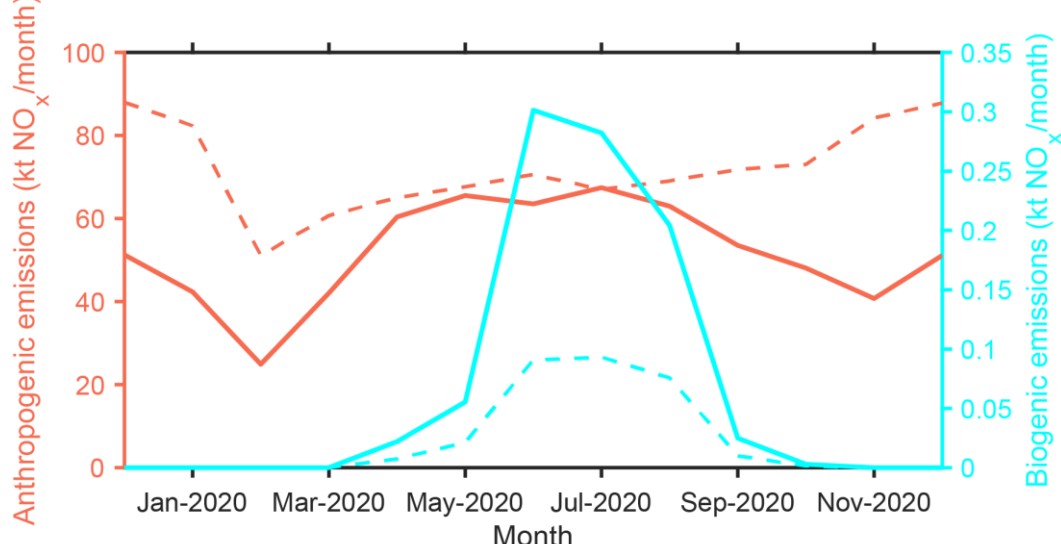

**Figure 6. Monthly NO$_x$ emissions from the grids dominated by anthropogenic (red) and biogenic (cyan) sources.** The
solid and dashed curves represent the posterior and the prior NO$_x$ emissions, respectively.





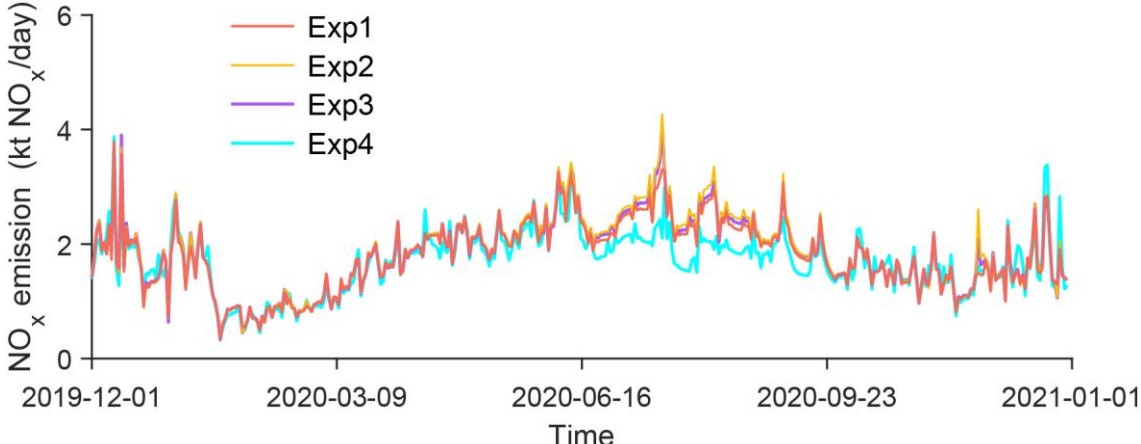

**Figure 7. The daily posterior emission time series from the sensitivity experiments Exp1-4 in the domain D03.**



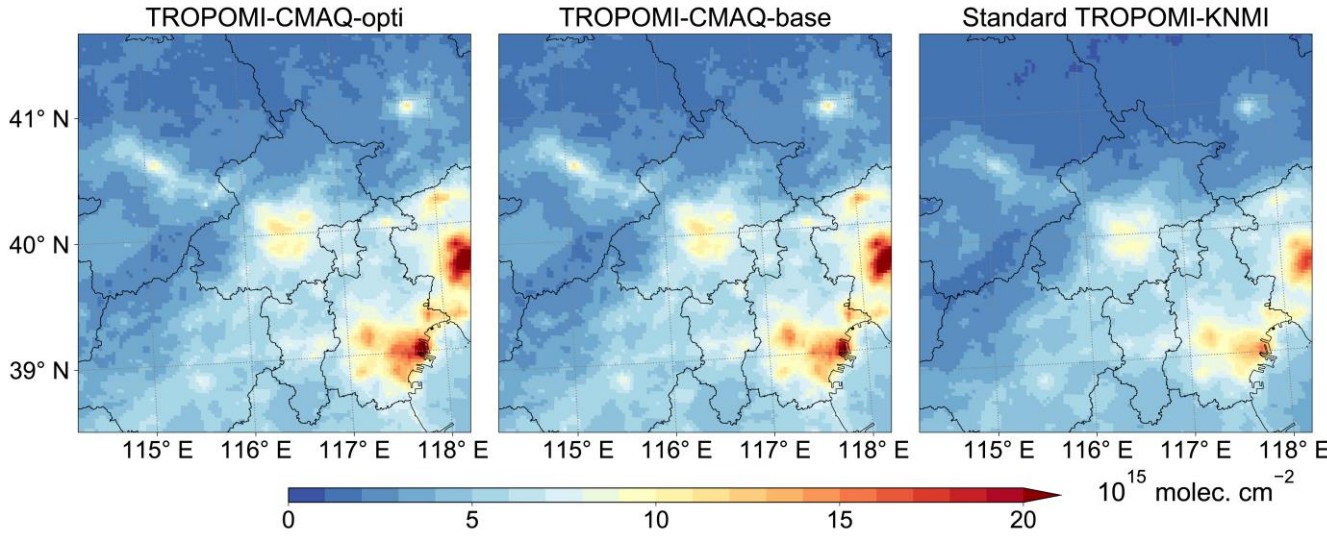

**Figure 8.** Comparison of the spatial distribution of the NO$_2$ TVCDs from the new TROPOMI NO$_2$ retrieval products and the standard TROPIMI-KNMI product in summer. TROPOMI-CMAQ-opti refers to the new TROPOMI product using the posterior CMAQ simulation as the a-priori profiles. TROPOMI-CMAQ-base refers to the new TROPOMI product using the prior CMAQ simulation as the a-priori profiles.





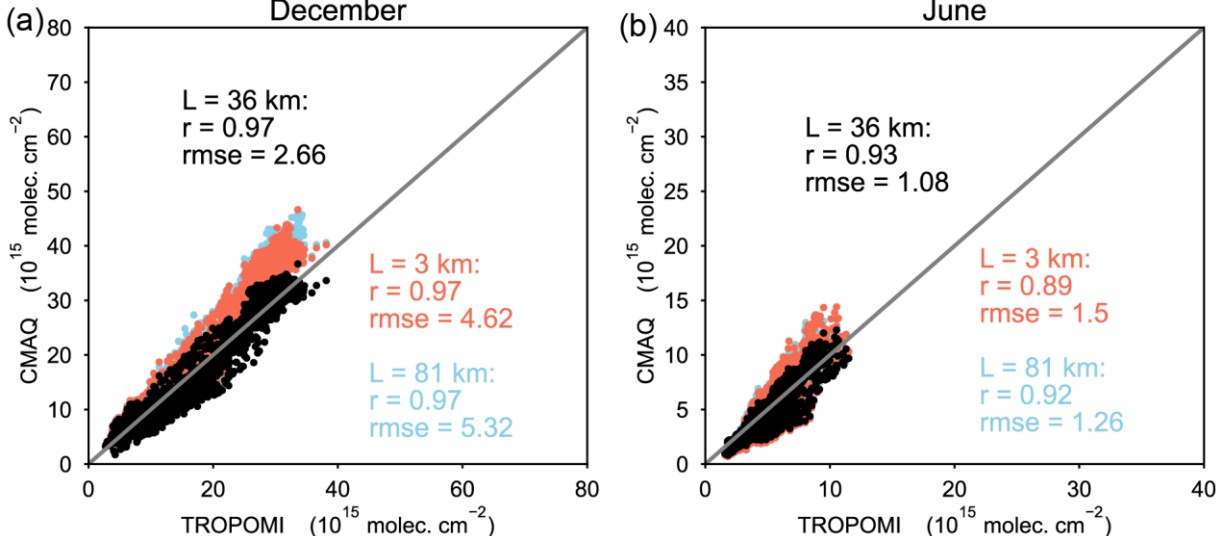

**Figure 9.** Scatter plots of the TROPOMI NO$_2$ TVCDs and the CMAQ simulated NO$_2$ TVCDs from the inversion experiments with a localization parameter of 3 km, 36km, and 81 km, respectively. (a) December 2019. (b) June 2020.