# Peer review of "Tracking daily NOx emissions from an urban agglomeration based on TROPOMI NO2 and a local ensemble transform Kalman filter"

_EGUsphere, 2024_

## Author Comment (AC1)

**Referee #1:**

This study estimates daily NO$_x$ emissions at a 3-km resolution in Beijing and its surrounding areas using an emission inversion framework. The framework assimilates TROPOMI NO$_2$ column concentrations with an Ensemble Kalman Filter coupled with CMAQ. The results reveal that proxy-based bottom-up emission datasets tend to overestimate NO$_x$ emissions in densely populated areas, providing crucial insights for urban air quality regulations. Robust sensitivity analyses further strengthen the study by evaluating the effects of satellite retrieval parameters (e.g., a priori profiles and averaging kernels) and an observation localization radius parameter on the inversion results. Specific comments on the manuscript are outlined below.

**Response:**

We thank the referee for the constructive and positive comments on our paper. We have provided our point-by-point responses as follows and revised the manuscript accordingly.

**Specific comments**

1. Figure 3 and Figure S4: What ground air quality monitoring station data is used for this comparison? Is it based on a single station or a multi-station average? Additionally, how does this comparison vary across different ground stations, such as those in densely populated areas versus suburban or rural areas?

**Response:**

Thanks for your valuable comments. We use the national control stations maintained by China National Environmental Monitoring Center (CNEMC) to evaluate the model simulations. Fig. S1 shows the spatial location of the stations. These stations are primarily located in densely populated areas of each city.

According to your suggestion, we have classified the observation stations into two categories based on NO$_2$ concentration characteristics, population density, and emission patterns: low-emission areas and high-emission areas. The low-pollution areas refer to the two northern cities in the D03 domain, Zhangjiakou and Chengde, where NO$_2$ concentrations are relatively low. The observation stations in all other urban areas are classified as high-pollution stations.

Fig. S6 presents a comparison between CMAQ simulations and ground-based observations in different regions. Compared to the highly polluted urban areas, the posterior NO$_2$ simulations in Zhangjiakou and Chengde show much better consistency with observations during summer. This indicates that in regions with low surface emissions, the accuracy of posterior simulations in summer is relatively high. Furthermore, it reinforces the finding that in high-emission urban areas, the constraint of satellite NO$_2$ column measurements on surface emissions in summer is weaker, leading to an overestimation in posterior simulations.

We have added a description of the air quality monitoring stations in Sect. 2.6 (Lines 262-263) as follows:

"Fig. S1 shows the spatial location of the ground-based stations used to evaluate the CMAQ simulations. These stations are primarily located in densely populated areas of each city."

We have added the analysis of the evaluation results for different regions in Sect. 3.1.2 (Lines 328-336) as follows:

"Furthermore, we have classified the observation stations into two categories based on NO₂ concentration characteristics, population density, and emission patterns: low-pollution areas and high-pollution areas. The low-pollution areas refer to the two northern cities in the D03 domain, Zhangjiakou and Chengde, where NO₂ concentrations are relatively low. The observation stations in all other urban areas are classified as high-pollution stations. Fig. S6 presents a comparison between CMAQ simulations and ground-based observations in different pollution regions. Compared to other highly polluted urban areas, the posterior NO₂ simulations in Zhangjiakou and Chengde show much better consistency with observations during summer. This indicates that in regions with low surface emissions, the accuracy of posterior simulations in summer is relatively high. Furthermore, it reinforces the finding that in highly polluted urban areas, the constraint of satellite NO₂ column measurements on surface emissions in summer is weaker, leading to an overestimation in posterior simulations."

[Figure]

Figure S6. Comparison of the ground-based daily (a) NO₂ and (b) O₃ concentrations with the CMAQ 3-km simulations utilizing prior and posterior NO$_x$ emissions as the model input. High-emission areas refer to the average of station measurements in the urban areas of Beijing, Tianjin, Baoding, Langfang, and Tangshan, while Low-emission areas refer to the average of station measurements in Zhangjiakou and Chengde.

2. Page 4, Lines 118-119: "The MEIC emission inventory is spatially and temporally allocated to match the CMAQ model domain using spatial proxies and empirical temporal profiles." What are the temporal and spatial resolutions of the MEIC inventory? What types of spatial proxies and temporal profiles are used to allocate emissions to the CMAQ model domain? Please elaborate further on these details in the paragraph.

Thanks for your good suggestion. We have added the details on the spatiotemporal allocation method for the MEIC emission inventory using spatial proxies and temporal profiles in Lines 126-132 (Sec. 2.1), as follows:

"The spatial proxies include total population density, urban population density, rural population density, and road length. These spatial proxies are updated annually to reflect interannual changes. The temporal profiles are unique for each major emission source. The

monthly profiles capture both seasonal variations and interannual trends in emissions, reflecting real activity levels. The allocation from monthly to daily values is achieved using sector-specific profiles that incorporate weekly and workday variations. In the CMAQ model, the MEIC inventory is mapped to the CMAQ model grids. Emissions from point sources are directly assigned to the grid cells where they are located, while emissions from area sources are first allocated to 1 km × 1 km grid cells based on the spatial proxies and then aggregated to the model grids based on WRF-CMAQ grid parameters."

3. Page 5, Lines 151-152: Does the inversion system presented in this study scale prior emissions on a daily scale? If so, how does the inversion system address hourly variations in $NO_x$ emissions? Does the inversion system adjust the hourly profiles of the bottom-up emission inventory? Please provide additional details on the time steps used for assimilating TROPOMI $NO_2$ data to scale prior emission inventories.

Thank you for your valuable comments. Yes, the inversion system presented in this study adjusts prior emissions on a daily scale. Due to the once-daily overpass of the TROPOMI satellite, we are unable to utilize TROPOMI observations to resolve hourly variations in $NO_x$ emissions. For hourly variations in $NO_x$ emissions, we still follow the daily variation pattern in the prior MEIC inventory, using hourly profiles for temporal emission allocation. The assimilation time step for scaling prior emissions using TROPOMI $NO_2$ data is set to one day. Specifically, the differences between satellite observations and model simulations (spatiotemporally collocated for afternoon overpasses) are used to update the daily scaling factors of emissions.

We have clarified this issue in Sec. 4.3 and discussed the future outlook of using GEMS satellite hourly observation data for hourly emission estimation (Lines 494-496 and Lines 499-502).

4. Page 9, Lines 249-250: I recommend including additional error metrics, such as mean percentage error, to further illustrate the improvement in posterior emissions simulations. This would help address the question, "In which season is the most significant improvement observed after inversion?"

Thanks for your good suggestion. We chose to use the percentage change in the root mean square error (RMSE) to quantify the improvement rate of posterior emissions relative to prior emissions, which provides a more intuitive understanding that the improvements are greater in the summer and autumn seasons compared to winter and spring. We made the following modifications in Sect. 3.1.1, Lines 284-286:

"Compared to the prior simulations, the RMSE differences between the posterior simulated $NO_2$ concentrations and the observations were reduced by 58.32%, 59.10%, 69.73%, and 70.03%, respectively."

5. Page 9, Lines 251-252: What spatial proxies are used in MEIC? For example, does it utilize road network shapefiles? Providing specific examples would make this argument more compelling and relevant.

The spatial proxies include total population density, urban population density, rural population density, and road length. We have clarified this in Lines 287 and 125-126.

6. Page 9, Lines 253-254: "However, the $NO_2$ TVCDs from prior simulations indicate substantial overestimations in urban environments across various seasons…" Please specify the seasons or months to provide clarification.

Done. We have clarified this in Line 296 (Sect. 3.1.1).

7. Page 10, Lines 282-286: Why doesn't the simulated $O_3$ concentration exhibit the "summer bias" that is clearly evident in the comparison between simulated $NO_2$ and observations? Please provide a more detailed discussion of the factors that could explain the differences between the simulations of $NO_2$ and $O_3$.

Thanks for your good suggestion. We have added the discussion of the factors that could explain the differences between the simulations of $NO_2$ and $O_3$ in Lines 340-347 (Sect. 3.1.2), as follows:

"The overestimation of bottom-up $NO_x$ emissions leads to negative biases in $O_3$ simulations throughout the year. However, during the summer, the deviations between simulated $O_3$ concentrations and observations are less pronounced compared to those observed in $NO_2$ simulations. This is because the simulation of $NO_2$ is significantly influenced by surface $NO_x$ emissions. In contrast, simulated $O_3$ concentrations are affected by multiple factors, with $NO_x$ emissions being only one of them. For instance, during summer, increased biogenic emissions and enhanced photochemical activity play critical roles. As a result, even in the prior $O_3$ simulations, the discrepancies between modeled and observed $O_3$ values are less pronounced compared to those in $NO_2$ simulations. Although the posterior $O_3$ simulations align well with observations, they remain slightly below the observed values, indicating that the posterior $NO_x$ emissions may still be overestimated during the summer."

8. Page 10, Lines 295-296: "However, the posterior emission maps substantially reduce emissions from city centers and reallocate these emissions to other areas, such as increasing emissions from inter-city transportation, among other changes." This is an important finding, but it requires more supporting evidence. From Figure 4 alone, it is challenging to identify the locations of inter-city transportation, making it difficult to confirm whether the reduced emissions from city centers are reallocated to road networks. Consider including an additional figure that overlays the locations of major inter-city roadways with the areas of increased emissions in the posterior estimates.

Thanks for your good suggestion. We have added a spatial distribution map of emissions for 2020 in the supplementary materials, overlaying road information on the third column of the figure (see Fig. S7). The roads include national highways, provincial roads, and expressways. Since these three types of roads cover the main arterial network, we did not display county and rural roads to ensure the clarity of the figure.

Combined with the difference map between posterior and prior emissions in the third column of Fig. 4, the results show reduced emissions in urban areas, while certain roads exhibit

significant emission increases. Notably, the emission increases along roads are most pronounced during spring and summer.

We have made revisions in Sect. 3.2.1 (Lines 356-359):

"However, the comparison between posterior and prior emissions (third column of Fig. 4 and Fig. S7) reveals that the posterior emission maps substantially reduce emissions in urban centers and redistribute these emissions to other regions. For instance, emissions from inter-city transportation (see the third column of Fig. S7 with the overlay of the road map) are notably increased, particularly during spring and summer."

[Figure]

Figure S7. The spatial map of the prior and posterior emissions in 2020. The last column displays the differences between the two emissions, overlaid with a road map that includes national highways, provincial roads, and expressways.

9. Page 10, Lines 301-302: "The posterior $NO_x$ emissions for the year 2020 (657 kt $NO_x$) decreased by 23.7% compared to the prior inventory (861 kt $NO_x$). The largest reductions occurred in winter and autumn, with declines of 44.5% and 36.4%, respectively." Does the bottom-up emission inventory (prior) account for the impact of COVID-19? If so, please provide an explanation in the paragraph.

Thanks for your good suggestion. We have clarified this in Lines 373-375 (Sect. 3.2.1)

"The bottom-up prior inventory is based on actual activity level data and emission factors, which can somewhat reflect the emission reductions during the COVID-19 lockdown period. However, due to statistical errors and the misrepresentation of emission factors, the prior emission inventory still fails to provide an accurate estimate of the regional total $NO_x$ emissions."

10. Page 11, Lines 319-321: "The seasonal variation of the posterior $NO_x$ emission estimate in our research is similar to the results obtained by previous studies (Wang et al., 2007; Qu et al., 2017; Miyazaki et al., 2017). Qu et al. (2017) utilized OMI measurements to infer the $NO_x$ emissions in China, and the seasonal pattern of $NO_x$ emissions for China and Beijing City is consistent with our study." How similar are these findings? Consider adding quantitative metrics to describe the seasonality of $NO_x$ emissions as observed in this study and in previous studies.

Sorry for not explaining it clearly. We found that in the studies listed (Wang et al., 2007; Qu et al., 2017; Miyazaki et al., 2017), the seasonal variation pattern of $NO_x$ emissions derived from satellite observations is similar to our study. Specifically, the posterior emissions based on satellite observations show significant seasonal variation, in contrast to the relatively insignificant seasonal changes in the prior emission inventories. Additionally, while the prior emissions tend to have the lowest levels in summer, the posterior emissions are higher in summer, comparable to or even higher than winter emissions (e.g., see Figures 12 and 14 in Qu et al., 2017). We compared the seasonal variation plots from these studies, but since the study areas differ in size, we did not perform a quantitative comparison of the changes in total regional emissions.

We have made modifications in Lines 412-414 (Sect. 3.2.2), as follows:

"These studies, along with the results of our study, indicate that while the prior emissions tend to have the lowest levels in summer, the posterior emissions are higher in summer, comparable to or even higher than winter emissions."

11. Page 12, Lines 364-365: "To evaluate the impact of different L values on the $NO_x$ emission inversions, we perform two additional experiments with L = 3 km (Exp_L3km) and L = 81 km (Exp_L81km), respectively." What are the reasons for choosing these specific L values, 3 km and 81 km, for the sensitivity analysis? Please provide an explanation.

Thanks for your good suggestion. Firstly, based on theoretical analysis, we initially determined that a localization radius of 36 km is appropriate. As elaborated in Section 2.4, the selection of this radius (36 km) was guided by the typical lifetime of $NO_2$ (~4 hours) and wind speed (~3 m/s) in the Beijing region (Wu et al., 2021) (see Lines 205-208 in our manuscript). However, this value serves only as a preliminary estimate, as the lifetime of $NO_2$ varies with environmental conditions and wind speed is not constant. To assess the sensitivity of emission inversion results to localization radius, we tested values smaller and larger than 36 km. We chose 3 km because the resolution of our model is 3 km, and this choice can be roughly understood as ignoring the effects of emission transport between grids. The choice of 81 km is based on the typical lifetime of $NO_2$ and wind speed, providing a relatively larger transport distance. Additionally, following the suggestion of Referee #2, we added another experiment with $L$ = 10 km to test whether the results would be better than those from the $L$ = 36 km experiment. We have made modifications in Sect. 4.2 (see Lines 466-481).

12. Figure 5: Consider adding a visual marker to highlight the implementation and relaxation of COVID-19 containment measures, as well as notable events such as the Chinese Lunar New Year holiday.

Thanks for your good suggestion. We have made revisions according to your comments as below (see Lines 380-386 and Fig. 5).

"The reduction of $NO_x$ emissions due to the pandemic lockdown measures lasted from early 2020 to mid-March 2020, with emission levels gradually returning to normal by late April. However, although the prior emission inventory partially reflects real monthly production activity levels, it fails to accurately capture such dynamic changes in emissions. The prior emission inventory only distinctly captures the emission reduction that occurred from early

February to mid-February 2020, mainly caused by the Chinese Lunar New Year. In addition, the posterior emission estimates also indicate a period of emission reduction and rebound from mid-June to mid-July 2020, coinciding with the sudden outbreak of the epidemic and the subsequent lockdown measures implemented at the Xinfadi market in Beijing, China."

[Figure]

**Figure 5.** Time series of the bottom-up and top-down daily $NO_x$ emissions for domain D03 and Beijing City. The gray dashed line indicates the Chinese Lunar New Year, which also marks the date when Beijing began implementing COVID-19 control measures. The blue dashed line represents the start date of control measures following the sudden outbreak at the Xinfadi market in Beijing. The gray shaded area represents the period affected by COVID-19 measures in 2020, and the light blue shaded area highlights the time frame impacted by the Xinfadi market outbreak.

13. Figure 5: It appears that the prior emission inventory exhibits a consistent diurnal cycle of hourly $NO_x$ emissions. How does this compare to the diurnal cycle in the posterior $NO_x$ emissions? Does the inversion system reveal a similar pattern? Please consider adding a figure and/or a paragraph to discuss this comparison.

Thank you for your valuable feedback and comments. Actually, Fig. 5 shows the time series of daily emissions, not hourly $NO_x$ emissions. The daily time series of the prior inventory shows periodic variations between weekdays and weekends, i.e., the MEIC inventory shows a clear weekly pattern. This is because, for the prior inventory, the allocation from monthly emissions to daily values is achieved using sector-specific profiles that incorporate weekly and workday variations. However, our satellite-derived emissions do not display a distinct weekly pattern. Previous studies on satellite observations and ground concentration variations have also indicated that such a weekly pattern is not prominent in China (Wei et al., 2022).

As for the MEIC inventory, it does include a diurnal cycle of hourly $NO_x$ emissions, as the MEIC inventory uses hourly profiles to allocate hourly emissions. However, since the TROPOMI satellite observations used in this study only provide afternoon overpasses per day, they do not support the derivation of hourly emission variation patterns. We have clarified this issure and added a discussion on the future aim to utilize GEMS hourly data to reveal the hourly variation patterns of emissions in Sect. 4.3 (Lines 494-496 and Lines 499-502).

**References**

Wang, Y., McElroy, M. B., Martin, R. V., Streets, D. G., Zhang, Q., and Fu, T.-M.: Seasonal variability of $NO_x$ emissions over east China constrained by satellite observations: Implications for combustion and microbial sources, Journal of Geophysical Research: Atmospheres, 112, https://doi.org/10.1029/2006JD007538, 2007.

Miyazaki, K., Eskes, H., Sudo, K., Boersma, K. F., Bowman, K., and Kanaya, Y.: Decadal changes in global surface $NO_x$ emissions from multi-constituent satellite data assimilation, Atmos. Chem. Phys., 17, 807-837, 10.5194/acp-17-807-2017, 2017.

Qu, Z., Henze, D. K., Capps, S. L., Wang, Y., Xu, X., Wang, J., and Keller, M.: Monthly top-down $NO_x$ emissions for China (2005–2012): A hybrid inversion method and trend analysis, Journal of Geophysical Research: Atmospheres, 122, 4600-4625, https://doi.org/10.1002/2016JD025852, 2017.

Wu, N., Geng, G. N., Yan, L., Bi, J. Z., Li, Y. S., Tong, D., Zheng, B., and Zhang, Q.: Improved spatial representation of a highly resolved emission inventory in China: evidence from TROPOMI measurements, Environ Res Lett, 16, ARTN 084056
10.1088/1748-9326/ac175f, 2021.

Wei, J., Liu, S., Li, Z., Liu, C., Qin, K., Liu, X., Pinker, R. T., Dickerson, R. R., Lin, J., Boersma, K. F., Sun, L., Li, R., Xue, W., Cui, Y., Zhang, C., and Wang, J.: Ground-Level $NO_2$ Surveillance from Space Across China for High Resolution Using Interpretable Spatiotemporally Weighted Artificial Intelligence, Environ. Sci. Technol., 56, 9988–9998, https://doi.org/10.1021/acs.est.2c03834, 2022

---

## Author Comment (AC2)

**Referee #2:**

In their paper, Yawen Kong and co-authors present the results of a $NO_x$ emission inversion system applied to the region around Beijing. It is one of the first emission assimilation systems targeting a high model (and emission) resolution of 3 km, matching well the TROPOMI footprint size and resolving the fine-scale information provided by the satellite. This inversion provides strong evidence that the proxies used to distribute the emissions in the MEIC inventory, especially the scaling with population density, has shortcomings. The paper is well written, has a good set of references and the figures provide a good documentation of the results. I am in favour of publishing these interesting results after my (relatively minor) comments have been dealt with and the answers have been incorporated in the text of the paper.

**Response:**

We thank the referee for the constructive and positive comments on our paper. We have provided our point-by-point responses as follows and revised the manuscript accordingly.

**Comments:**

I l 113: "simulations over the first and second domains were performed before the inversion experiments to provide boundary conditions for the third domain." Because the emissions in D02 are not adjusted, this may lead to inconsistencies inside/near the boundary of D03 and sub-optimal emission estimates. Please comment or refer to a later discussion of this nesting feature.

**Response:**

Thanks for your valuable comments. We agree that using unoptimized emissions from D01 and D02 to simulate boundary conditions for D03 could introduce uncertainties in the emission inversion, particularly near the domain boundaries. These uncertainties are more pronounced in winter due to the longer $NO_2$ lifetime and stronger influence of atmospheric transport. As shown in Fig. 2, the simulated concentrations near the southern boundary of D03 in autumn and winter are higher than observations, likely caused by unadjusted D02 boundary conditions. This discrepancy may lead to underestimated emission inversion results in these regions. In contrast, during the spring and summer, the influence of the D02 boundary field on D03 concentrations is weaker, and the impact on emission inversion is smaller.

Theoretically, optimizing emissions for the D02 region and using the posterior concentrations as boundary conditions for D03 could reduce such errors. However, due to the high computational demands of high-resolution $NO_2$ chemical transport modeling and assimilating dense TROPOMI satellite data, we omitted the optimization of outer domains (D01/D02) in this study. If computational resources are sufficient, we recommend optimizing emissions in D02 to reduce the influence of boundary conditions on D03. Alternatively, we could consider expanding the D03 domain by extending its boundaries outward by approximately 100 km (the distance potentially influenced by boundary conditions). After completing the assimilation experiments for D03, the extended area could be trimmed, retaining only the central region that is minimally affected by boundary conditions for $NO_x$ emission analysis.

We have added a brief explanation of this issue in Lines 115-119 (Sect.2.1) as follows:

"In principle, performing assimilation inversion for the D02 region first, and using the posterior simulated concentrations as boundary conditions for the D03 region, would reduce boundary field errors and improve emission inversion results near the D03 boundary. However, in this study, we made a trade-off between minimizing computational resources and optimizing the boundary field, ultimately performing assimilation inversion only for the D03 region."

Additionally, we have added a discussion in Section 4.3 to address this issue (Lines 506-535):

"In our model assimilation system, using unoptimized emissions from D01 and D02 to simulate boundary conditions might introduce uncertainties in the emission inversion near the D03 boundaries. These uncertainties are more pronounced in winter due to the longer $NO_2$ lifetime and stronger atmospheric transport effects. As shown in Figure 2, simulated concentrations near the southern boundary of D03 in autumn and winter are higher than observations, likely due to the influence of unadjusted D02 boundary conditions, which may lead to underestimated emission inversion results in these regions. In contrast, the impact of D02 boundary conditions on D03 is weaker in spring and summer. If computational resources are sufficient, we recommend optimizing emissions in D02 to reduce the influence of boundary conditions on D03. Alternatively, we could consider expanding the D03 domain by extending its boundaries outward by approximately 100 km (the distance potentially influenced by boundary conditions). After completing the assimilation experiments for D03, the extended area could be trimmed, retaining only the central region that is minimally affected by boundary conditions for $NO_x$ emission analysis."

2. l 116: Please provide the spatial resolution of the MEIC inventory.

The spatial resolution of the MEIC inventory is the same with the WRF-CMAQ model configured in this study. We apologize for not clearly explaining the spatiotemporal allocation and resolution of the MEIC inventory. In response to Comments 2-4, we provided additional details in Lines 125-132 to elaborate on the spatiotemporal allocation of the MEIC inventory in this study, as follows:

"In this study, the MEIC emission inventory is spatially and temporally allocated to match the CMAQ model grid cells using spatial proxies and empirical temporal profiles. The spatial proxies include total population density, urban population density, rural population density, and road length. These spatial proxies are updated annually to reflect interannual changes. The temporal profiles are unique for each major emission source. The monthly profiles capture both seasonal variations and interannual trends in emissions, reflecting real activity levels. The allocation from monthly to daily values is achieved using sector-specific profiles that incorporate weekly and workday variations. In the CMAQ model, the MEIC inventory is mapped to the CMAQ model grids. Emissions from point sources are directly assigned to the grid cells where they are located, while emissions from area sources are first allocated to 1 km × 1 km grid cells based on the spatial proxies and then aggregated to the model grids based on WRF-CMAQ grid parameters."

3. l 116: Please elaborate on the temporal profiles in MEIC: does this include

sector-dependent seasonal, weekday and diurnal patterns? Are these considered to be realistic?

The temporal profiles are designed based on the MEIC Level 4 source classification, with each major emission source category assigned unique temporal profiles. The monthly profiles capture both seasonal variations and interannual trends in emissions, reflecting real activity levels. The allocation from monthly to daily values is achieved using sector-specific profiles that incorporate weekly and workday variations. We have added a description to elaborate on the spatiotemporal allocation of the MEIC inventory in this study (see Lines 126-129).

4. l 119: "inventory is spatially and temporally allocated to match the CMAQ model domain" Is the grid of CMAQ adjusted to match MEIC, e.g. 3x3 gridcells of MEIC in one gridcell of CMAQ?

In the CMAQ model, the MEIC inventory is allocated to the CMAQ model grids. Emissions from point sources are directly assigned to the grid cells where they are located, while emission from area sources are first allocated to the 1 km $\times$ 1 km grid cells based on the spatial proxies and then aggregated to the model grids based on WRF-CMAQ grid parameters. We have added additional details to elaborate on the spatiotemporal allocation of the MEIC inventory in this study (see Lines 129-132).

5. l 133: The TROPOMI reprocessing with processor version 2.4 is now available for several years. Why did the authors use 2.3.1?

Thanks for pointing out this problem. In this study, we adopted version 2.3.1 because this work was initiated earlier, and during the development and debugging of our model and assimilation system, we found that version 2.3.1 was the latest and recommended version at that time. For future work, our system will be updated to support the latest version of the TROPOMI data product. We have added the following clarification in Lines 145-147:

"Since the updated TROPOMI data products (version 2.4.0 and subsequent versions) have become available, our system will be updated in future work to support the latest versions of the TROPOMI data product."

6. Section 2.2: Please discuss also the satellite retrieval uncertainties. Did you use the uncertainty from the L2 file to compute the covariances?

For the observation error covariances of the satellite data, we adopted the uncertainties provided in the satellite Level 2 (L2) data products. However, since the original L2 files contain multiple L2 orbital files per day, and there may be overlaps between orbits, we preprocess and merge them into a single daily data file. Consequently, the corresponding uncertainties come from the processed data file.

First, we map the data from individual L2 files onto the CMAQ D03 model grid using an area-weighted method. Then, we merge them into daily files. This processing is applied to all variables used in this study, including tropospheric $NO_2$ column concentrations, retrieval uncertainties, and averaging kernels.

We have modified the descriptions in Lines 152-155:

"For comparison with the CMAQ model, we used the area-weighted average method to convert the satellite Level 2 (L2) orbital files into a gridded data product to match the resolution of the CMAQ D03 model grid cell (3 km×3 km). This processing is applied to all variables used in this study, including tropospheric $NO_2$ column concentrations, retrieval uncertainties, and averaging kernels."

7. l 136: " quality assurance value greater than 0.5" and "cloud fraction exceeding 40%". This is not following the default recommendation, which is a value greater than 0.75 for most applications. Please motivate why you deviate from the standard filtering?

Thank you for raising this point. Our choice of a QA value threshold > 0.5 aligns with the recommendations in Eskes et al. (2022). While the manual suggests a default threshold of > 0.75 for most users, it explicitly states that a threshold of > 0.5 retains additional high-quality retrievals over clouds and snow/ice (while still filtering errors) and is particularly suitable for assimilation and model comparison studies utilizing averaging kernels. Since our study focuses on these objectives, we adopted the lower threshold to maximize the availability of reliable data for robust analysis.

We have made revisions in Lines 148-151 as follows:

"We first selected the pixels with a quality assurance value greater than 0.5, as recommended by the product user manual (Eskes et al., 2022), to ensure data quality and maximize the availability of high-quality observations for assimilation and model comparison purposes through the use of averaging kernels within the observation operator."

8. l 135: It would be good to mention here that the averaging kernels are also used in the observation operator.

Thanks for this good suggestion. We have made revisions in Line 151 as follows:

"We first selected the pixels with a quality assurance value greater than 0.5, as recommended by the product user manual (Eskes et al., 2022), to ensure data quality and maximize the availability of high-quality observations for assimilation and model comparison purposes through the use of averaging kernels within the observation operator."

9. l 139: Gridded satellite product on 3x3 km grid? I get the impression that the satellite data is first mapped onto the model grid of 3x3km, and that subsequently these grid observations are assimilated. Is this true? The normal procedure would be to (area) average the model over the footprint of the satellite. Is there a reason why this approach was chosen?

Since we need to merge different L2 orbital data files, and these orbital data may have overlaps, we first map them onto a common grid coordinate system before merging them into a daily data file. For this study, it is natural to choose the 3 km model grid as the coordinate system. Additionally, this approach facilitates a direct comparison between the model-simulated column concentrations and the satellite-derived column concentrations, as shown in Figure 2.

10. l 151: "optimizing the initial $NO_2$ concentrations by assimilating the satellite observations has minimal impact ". Please refer to Miyazaki, who is adjusting both concentrations and

emissions. Would there be an advantage (disadvantage) of using such a mixed concentration and emission state vector?

Thanks for your valuable comments. In our opinion, whether adjusting both concentrations and emissions using a mixed concentration and emission state vector has an advantage depends on the assimilation frequency and the assimilation-simulation setup. If the observational data are available hourly, then adjusting both concentrations and emissions at each time step could theoretically improve the model's initial concentrations, thereby reducing the impact of initial concentration errors on subsequent emission inversion. However, in our study, TROPOMI data are available in the local afternoon, while our model simulation initializes at 00:00 each day. Therefore, under our assimilation setup, improving the initial concentration may not yield significant benefits. However, the potential advantages of simultaneously optimizing both concentrations and emissions would need to be further investigated through experimental testing and comparative studies.

11. l 150: x is an emission scaling factor. Please state that x=1 means that the emission is equal to the MEIC emission at a given location and time. This will be helpful for the reader.

Thanks for this good suggestion. We have added the following statement in Line 170:

"When the scaling factors are 1, it means that the emissions are equal to the MEIC emissions at a given location and time."

12. l 154: What is the assimilation time step? Is it one day?   (t - 1 = t - 1 day?)

Yes, the assimilation time step is one day.

13. Eq 1: In line 149, x has an ensemble member subscript. In Equation 1 the subscript refers to time. Please improve.

Thanks for this good suggestion. We have clarified the relationship between $\overline{x}$ and $x_i$ in Equation 1 through the following revisions:

"$\overline{x}$ is the ensemble mean of the state vector $x_i$ ($i = 1,2,…,k$)."

14. Eq. 1: Please explain this choice in more detail. Why is the average over two days (and not e.g. only the last day) Why is the "+1" term added, which relaxes back to the prior emission inventory? Why is persistency (x_t = x_{t-1}) not a better choice?

If we do not consider the previous assimilation information, Equation 1 would take the form: $\overline{x}_t^b = 1$, meaning that the emissions at time $t$ are directly based on the MEIC inventory at time $t$. To incorporate the optimized information from the observations of the two previous time steps while preserving the emission characteristics of the MEIC inventory at time $t$ (which is the role of the "+1" term), we adopted the form of Equation 1, following the approach of Peters et al. (2007) and our previous study (Kong et al., 2022). In contrast, the form $\overline{x}_t^b = \overline{x}_{t-1}^a$ only considers the optimized information from the previous time step, which is less temporally smooth and does not account for the emission characteristics at time $t$.

We have made modifications in Lines 179-181 as follows:

"With the dynamic model, the optimized information from the two previous time steps can be propagated to the current state, effectively implementing a moving average smoothing technique to reduce fluctuations in $\overline{x}^b$ over time."

15. l 179: "the prior error covariance maintains a fixed uncertainty value to prevent filter divergence, which is similar to our previous study (Kong 2022)" Please provide more details here.

We apologize for not describing this point clearly. We have revised the description in Lines 199-202 as below:

"The prior covariance matrix is constructed based on a normal distribution, with the standard deviation for the prior emissions is prescribed as 100%. The ensemble perturbation matrix $\mathbf{X}^b$ was constructed through Cholesky decomposition to the prior covariance matrix. To prevent filter divergence, the fixed prior error covariance structure is used in every assimilation cycle, which is similar to our previous study (Kong et al., 2022)."

16. l 186: Sensitivity experiments: The choices 3 and 81 km are very far apart. For me it would be more logical to test also 10 km and see if this improves the analysis.

Thanks for this good suggestion. We have conducted an additional set of experiments to test the improvement in emission inversion results using a localization parameter ($L$) of 10 km. The results demonstrate that the experiment with a 36 km localization parameter (Exp_L36km) still performs best overall. The inversion results from Exp_L10km are highly comparable to those of Exp_L36km, particularly showing better performance than Exp_L3km and Exp_L81km in winter. In summer, while the statistical metrics of Exp_L10km are slightly weaker than those of Exp_L81km, the posterior simulations in high-value regions show better consistency with observations compared to Exp_L3km and Exp_L81km. Testing the 10 km parameter is meaningful: if the optimization results with $L = 10$ km surpass those of $L = 36$ km, it would imply that fewer observations need to be assimilated, thereby reducing computational resource consumption. This is because the LETKF assimilation algorithm employs explicit localization, meaning only observations within the localization radius centered on each grid cell are assimilated.

We have made revisions in Sect. 2.5 to add an additional experiment to test the improvement in emission inversion results using a localization parameter ($L$) of 10 km (see Table 1). We have revised the discussion of the experimental results (see Fig. 9) in Sect. 4.2, as follows:

"The results showed that the three additional sensitivity experiments with $L$ values of 3 km, 10 km, and 81 km were all able to optimize $NO_x$ emissions to some extent. However, Exp_L36km still performed best overall. In winter, the effectiveness was the weakest for Exp_L81km, followed by Exp_L3km. The inversion results from Exp_L10km were closely comparable to those of Exp_L36km, though the statistical metrics were slightly lower than those of Exp_L36km. In summer, while the statistical metrics of Exp_L10km were slightly weaker than those of Exp_L81km, the posterior simulations in high-value regions showed better consistency with observations compared to Exp_L3km and Exp_L81km. Exp_L81km exhibited greater dispersion relative to observations, possibly because the 81 km × 81 km area included a large number of observations, which could interfere with the optimization of

emissions in the target grid. On the other hand, observations at longer distances were already outside the influence range of emission sources from the target grid. The 3 km radius area contained too few observations and has insufficient responsiveness to the transport of emissions from the target grid. Testing the 10 km localization parameter is meaningful: if the optimization results with $L$ = 10 km surpass those with $L$ = 36 km, it would suggest that fewer observations need to be assimilated, thereby reducing computational resource consumption. This is because the LETKF assimilation algorithm employs explicit localization, meaning that only observations within the localization radius centered on each grid cell are assimilated.

Since Exp_L36km yielded the best results in this study, using a 36 km localization radius is reasonable. In practice, selecting an appropriate localization parameter may require extensive tuning or adaptive adjustments based on meteorological conditions, but this would impose a significant computational burden. Therefore, we prefer to choose a theoretically reasonable localization parameter or observation localization strategy."

[Figure]

**Figure 9.** Scatter plots of the TROPOMI NO$_2$ TVCDs and the CMAQ simulated NO$_2$ TVCDs from the inversion experiments with a localization parameter of 3 km, 10 km, 36 km, and 81 km, respectively. (a) December 2019. (b) June 2020.

17. l 187: "exclude the days with satellite coverage below 70%". Why is this needed? If part of the domain is cloud-free, the observations in this part could lead to useful constraints for emissions.

We removed days with satellite coverage below 70% and did not use the remaining observations to optimize emissions for those days primarily for the following reasons. When satellite observations are significantly lacking on a given day, it usually means that there are large continuous spatial gaps in the observation data. As a result, emissions in these areas cannot be optimized, leading to emission estimates in those regions remaining equal to the original MEIC emissions. This would introduce significant interference in tracking and analyzing the daily variations in total regional emissions. We observed that data gaps rarely persist over multiple consecutive days. Therefore, for days with substantial data gaps, we used interpolated emissions from the nearest available days before and after to represent the emissions for those days and analyze the characteristics of daily variations in total regional emissions.

18. 1 194: "retrievals are resampled to model 3km grid." (related to my earlier comment) Normally model values are resampled to the satellite footprint in the observation operator to represent individual observations. The procedure is not very clear to me. Please provide details and motivate why a resampling of observations was done. Are the kernels resampled in the same way?

The original L2 files contain multiple L2 orbital files per day, and there may be overlaps between orbits. Therefore, we need to map them onto a common grid coordinate system before merging them into a daily data file. For this study, it is natural to choose the 3 km model grid as the coordinate system.

First, we map the data from individual L2 orbital files onto the CMAQ D03 model grid using an area-weighted method. Then, we merge them into daily files. This processing is applied to all variables used in this study, including tropospheric $NO_2$ column concentrations, retrieval uncertainties, and averaging kernels. We have clarified this issure in Lines 151-155.

19. Section 2.6: The equipment used to measure $NO_2$ for air quality monitoring purposes is known to be influenced by other nitrogen oxides like PAN or $HNO_3$. In previous studies, like e.g. Lamsal et al., 2008, doi:10.1029/2007JD009235, the comparisons with ground-based observations have been done by including these other species in the comparison, see e.g. eq.1 in this paper. What about the Chinese surface measurements? Are they also based on molybdenum converter instruments and do they have a similar problem? If so, I would propose to mention the issue and possibly correct for it in the comparison with CMAQ.

Thanks for pointing out this issue. Yes, the measurements of $NO_2$ concentrations are obtained via a chemiluminescence analyzer equipped with a molybdenum converter, which is known to be interfered with by other nitrogen oxides, such as PAN or $HNO_3$. As noted in Lamsal et al. (2008), $NO_2$ concentrations derived from observational instruments may be overestimated, and a correction factor ($\leq 1$) was proposed to address this bias. However, in our study, we did not apply such corrections to the observational data, as our simulations do not output other reactive oxidized nitrogen species in our model, which are required for calculating the correction factor.

To evaluate the model performance, we compared both prior and posterior simulations with observations (see Fig. S4 for the D01 region and Fig. 3 for the D03 region). For the D01 region, prior simulations overestimated $NO_2$ concentrations during early 2020 and summer compared to observations. For the D03 region, prior simulations exhibited year-round overestimation relative to ground-based $NO_2$ observations, while posterior simulations significantly reduced discrepancies, though a positive bias persisted in summer. Importantly, even if the observed $NO_2$ values were adjusted by a correction factor ($<1$), the relative improvements between prior and posterior simulations would remain valid. This robustly demonstrates that the posterior emissions improved the CMAQ $NO_2$ simulation performance.

We clarified this issue in the revised manuscript (Lines 265-270 and Lines 324-327)

Lines 265-270:

"The measurements of $NO_2$ concentrations are obtained via a chemiluminescence analyzer equipped with a molybdenum converter, which is known to be interfered with by other

nitrogen oxides, such as PAN or HNO₃. As noted in Lamsal et al. (2008), NO₂ concentrations derived from observational instruments may be overestimated, and a correction factor (≤1) was proposed to address this bias. In this study, we did not apply corrections to the observations, and we discuss this point in Sect. 3.1.2. The systematic errors in ground-based NO₂ observations do not affect our conclusions, i.e., our posterior emissions have improved the CMAQ NO₂ simulation performance."

Lines 324-327:

"It is worth mentioning that measurements of NO₂ concentrations via the molybdenum converter may be overestimated due to interference from other reactive oxidized nitrogen species. Lamsal et al. (2008) proposed a correction factor less than 1 for NO₂ observation data. In this study, NO₂ observations are lower than prior simulations; therefore, applying such corrections to the NO₂ observations would not affect the conclusions of our study."

20. Fig.2. I assume that the kernels have been used in this comparison (e.g. Exp 1 in Table 1). Please mention this in the caption.

Done.

21. l. 267: "Our inversions at the 3 km scale were limited to optimizing emissions in the innermost domain and did not address emissions outside this area." The high-resolution domain is quite small (350km) and especially the edges of the domain will be influenced by the coarser-resolution middle domain. Transport of NO$_x$ from the source can cover 50-100km. As future improvement it may be useful to apply the assimilation also to the D02 domain for a better consistency.

Thanks for your valuable suggestion. We have added a discussion in Sect. 4.3 on potential future improvement (see Lines 506-535).

22. l 276: "during the summer was relatively limited". I was wondering if the results may have been influenced by the free tropospheric column? Did the authors check if free-tropospheric NO₂ concentrations and profile shapes in CMAQ are reasonable, especially in Summer? Are there e.g. aircraft profiles available? Alternatively, the profiles (Fig. S5) may be compared by other modelling results, e.g. with the CAMS simulations (as described in the Inness paper). Deep convection and lightning are important sources in Summer.

Thanks for your valuable comments. We did not employ observational profile data for validation, and studies on mid- to upper-tropospheric profiles in this region remain relatively scarce. For comparison with the CAMS simulations, we referenced only the results presented in Douros et al. (2023), which focused on Paris. In Sect. 4.3, we have added a discussions on this limitation and recommend that future studies incorporate independent observational validation or comparisons with other models (e.g., CAMS) evaluate the impacts of summer lightning and deep convection processes (see Lines 536-539).

23. Sect. 3.2.2. MEIC shows a clear weekly pattern. This is not so clear in the satellite derived emissions. Please add a remark on this. I was wondering if the emission time averaging (Eq 1) is removing most of the weekly cycle. Or is there still a weekly cycle signal available in the posterior emissions?

The MEIC inventory exhibits a weekly pattern because its temporal distribution employs a profile that varies with weekdays. Our satellite-derived emissions do not display a distinct weekly pattern. Previous studies on satellite observations and ground concentration variations have also indicated that such a weekly pattern is not prominent in China (Wei et al., 2022). It is considered that the operation in Eq. 1 does not affect this weekly variation mode, as it is merely used to construct the prior emission estimates. The system further refines these emission estimates by assimilating satellite observations to derive posterior emissions through inversion, ensuring that the final results remain consistent with real-world variability.

24. l 329: "the prior biogenic emissions are significantly underestimated". Fig. 6 is interesting and shows a clear enhancement of the biogenic part in the assimilation. But these results may be partly misleading. The problems with the spatial disaggregation of MEIC anthropogenic emissions (too much following the population density) may be partly compensated by increases in biogenic $NO_x$ (which are more pronounced outside the populated areas). Furthermore, the biogenic emissions for this D3 region are much smaller than the anthropogenic emissions, as shown by Fig. 6. So, to my opinion the conclusion that biogenic emissions are underestimated should be formulated carefully. Are there references which support the statement that biogenic (soil) emissions are underestimated (in China)?

Thanks for your valuable comments. Previous bottom-up studies have suggested an underestimation of soil $NO_x$ emissions in the North China Plain during summer (Lu et al., 2021), but these findings are regionally and methodologically inconsistent with our study and therefore cannot support our conclusions. Although Fig. 6 indicates an increase in posterior biogenic emissions, the satellite observations used in our top-down approach integrate concentrations influenced by both anthropogenic and natural sources. This inherent limitation makes it challenging to isolate biogenic emissions with high confidence. Specifically, we cannot definitively state that grids labeled as "biogenically dominated" in the prior inventory truly represent pure biogenic contributions. Consequently, we have removed this conclusion from the manuscript.

25. l 350: "it may not be necessary to update the a priori profiles in each assimilation step". These are nice and convincing results, presented in Figs S7 and 8. Because domain D03 is such a high emission domain, the profile shape is mainly resulting from the emissions from the surface. But perhaps the results may be different for other regions with lower anthropogenic emissions, where the free-tropospheric relative contribution to the satellite-observed column may be larger, increasing the dependency of the profile shape to changes in surface emissions.

Thank you for this good suggestion. We have added discussions in Lines 455-458 based on your comments:

"Since the region D03 in our study is a high-emission area, its profile shape is mainly dominated by surface emissions. For regions with lower anthropogenic emissions, the free troposphere's contribution to the satellite-observed column is likely more pronounced. This enhanced contribution increases the profile shape's sensitivity to surface emission changes, which may result in significantly different inversion results compared to high-emission regions."

26. Sect. 4.2: The choice of the localization length is also a trade-off between ensemble size and "emission noise" resulting from the limited number of ensemble members. Are there indications of spurious emission increments in the 81 km experiment?

Thank you for raising this point. While we did not conduct a targeted analysis of the spurious emission increments, our analysis suggests that simulated observations within the 81 km radius may generate spurious correlations with emission perturbations in the central grid, leading to worse results compared to the $L$ = 36 km experiment.

27. Sect. 4.2: The step between 3 and 36 km is very large. For me it would have been logical to include also a L=10 km experiment.

Done. We have added the $L$=10 km experiment.

28. Sect. 4.2: I was missing a discussion about covariance inflation, which is normally needed to avoid a collapse and too much trust in the analysis. But I understand (line 179) that the prior error covariance maintains a fixed value to avoid filter divergence. Please provide the details: what is the fixed value used, and how is this implemented?

Done. We have added more details on the implementation in Sect. 2.4 (Lines 199-202).

29. l 380: "min-afternoon" ?

Thanks for pointing out this error. We have revised it to "mid-afternoon" .

30. Sect. 4.3. The GEMS geostationary observations are not mentioned. Is it considered to make use of GEMS hourly data in the future?

Thanks for your good suggestion. We have added a discussion on the future aims to make use of GEMS hourly data in Sect. 4.3 (Lines 499-502).

31. Section 4.3 / conclusions:
Some extra discussion on factors influencing the derived emissions would be useful. Emissions are clearly not the only uncertain factor in the model. The lifetime of $NO_2$ in the atmosphere is a key factor to relate satellite-observed concentrations to emissions, and errors in this could introduce a seasonally-dependent bias. The profile shape was discussed well, but was not really validated/verified with independent observations or modelling results. Natural emissions and concentrations in the free troposphere may not be very important for the D03 high emission region, but could be important elsewhere (It is mentioned that lightning is not included). And also the satellite retrievals have systematic uncertainties which could also introduce a seasonality.

Thanks for your valuable suggestions. We have revised the discussion in Sect. 4.3 as below:

"The uncertainties in the $NO_x$ emission inversion in this study arise from multiple sources, including inaccurate depictions of the chemical and physical mechanisms in the CTMs, methodological challenges in matching model simulations to satellite column concentrations, errors inherent to satellite retrievals, and limitations of current assimilation techniques, among others. The lifetime of $NO_2$ in the atmosphere is a key factor to relate satellite-observed concentrations to emissions, and errors in this could introduce a seasonally-dependent bias.

In terms of our emission inversion framework, several factors can introduce uncertainties to the inversions. Firstly, the settings of the CTM simulations, such as the configuration of the model layers, might be important for the depiction of the column concentrations where future improvements can be made. Secondly, the TROPOMI satellite's local overpass time in the min-afternoon means that the inversion system still cannot capture hourly variations in $NO_x$ emissions, and therefore the hourly allocations in the posterior emissions are the same as in the bottom-up inventory. Besides, the weak sensitivity of the satellite measurements to the $NO_2$ concentrations at lower altitudes leads to limited constraints on surface emissions, particularly during the summer. A recent study by He et al. (2022) suggests that the incorporation of hourly ground-based observations may help to improve surface emission inversions. The Geostationary Environment Monitoring Spectrometer (GEMS), launched in 2020, monitors atmospheric pollutants over Asia from a geostationary orbit, providing hourly observation data. Future work will involve assimilating GEMS satellite data, which will not only help track daily emission changes but also reveal the hourly variation patterns of emissions. In addition, although data assimilation methods take into account the influence of the observations outside of the target grid, the optimal influence distance is difficult to determine. An improved observation assimilation scheme, such as taking into account the correlation between the emissions in the target grid with the observations within the localization area may hold promise for improving the accuracy of emissions inversions.

Finally, our study lacks independent validation of the vertical $NO_2$ profile simulations to assess the relative contribution of free tropospheric activity to satellite-observed column concentrations under varying emission scenarios and to analyze the sensitivity of profile shapes to surface emission changes. Future work will incorporate observational validation and inter-model comparisons to address these limitations."

**References**

Eskes, H. J., van Geffen, J. H. G. M., Boersma, K. F., Eichmann, K.-U., Apituley, A., Pedergnana, M., Sneep, M., Veefkind, J. P., and Loyola, D.: Sentinel-5 precursor/TROPOMI Level 2 Product User Manual Nitrogendioxide, Report S5P-KNMI-L2-0021-MA, version 4.1.0, ESA, https://sentinel.esa.int/web/sentinel/technical-guides/sentinel-5p/products-algorithms/ (last access: 7 October 2023), 2022.

Peters, W., Jacobson, A. R., Sweeney, C., Andrews, A. E., Conway, T. J., Masarie, K., Miller, J. B., Bruhwiler, L. M. P., Pétron, G., Hirsch, A. I., Worthy, D. E. J., van der Werf, G. R., Randerson, J. T., Wennberg, P. O., Krol, M. C., and Tans, P. P.: An atmospheric perspective on North American carbon dioxide exchange: CarbonTracker, Proc. Natl. Acad. Sci. U.S.A., 104, 18925-18930, 10.1073/pnas.0708986104, 2007.

Kong, Y., Zheng, B., Zhang, Q., and He, K.: Global and regional carbon budget for 2015–2020 inferred from OCO-2 based on an ensemble Kalman filter coupled with GEOS-Chem, Atmos. Chem. Phys., 22, 10769-10788, 10.5194/acp-22-10769-2022, 2022.

Lamsal, L. N., R. V. Martin, A. van Donkelaar, M. Steinbacher, E. A. Celarier, E. Bucsela, E. J. Dunlea, and J. P. Pinto (2008), Ground-level nitrogen dioxide concentrations inferred from the satellite-borne Ozone Monitoring Instrument, J. Geophys. Res., 113, D16308, doi:10.1029/2007JD009235.

Wei, J., Liu, S., Li, Z., Liu, C., Qin, K., Liu, X., Pinker, R. T., Dickerson, R. R., Lin, J., Boersma, K. F., Sun, L., Li, R., Xue, W., Cui, Y., Zhang, C., and Wang, J.: Ground-Level $NO_2$ Surveillance from Space Across China for High Resolution Using Interpretable Spatiotemporally Weighted Artificial Intelligence, Environ. Sci. Technol., 56, 9988–9998, https://doi.org/10.1021/acs.est.2c03834, 2022

Lu, X., Ye, X., Zhou, M. *et al.* The underappreciated role of agricultural soil nitrogen oxide emissions in ozone pollution regulation in North China. *Nat Commun* **12**, 5021 (2021). https://doi.org/10.1038/s41467-021-25147-9